# MAGE: Multi-scale Autoregressive Generation for Offline Reinforcement Learning

**Chenxing Lin**[ab*], **Xinhui Gao**[ab*], **Haipen Zhang**[ab], **Xinran Li**[c], **Haitao Wang**[ab],
**Songzhu Mei**[d], **Chenglu Wen**[ab], **Weiquan Liu**[abe], **Siqi Shen**[ab†] **Cheng Wang**[ab]

[a]Fujian Key Laboratory of Urban Intelligent Sensing and Computing,
School of Informatics, Xiamen University (XMU), China
[b]Key Laboratory of Multimedia Trusted Perception and Efficient Computing, XMU, China
[c]Hong Kong University of Science and Technology
[d]School of Computer, National University of Defense Technology, China
[e]College of Computer Engineering, Jimei University, China
{lincx1123,xinhuigao,zhanghaipeng,haitaowang}@stu.xmu.edu.cn,
{siqishen,cwang,clwen,wqliu}@xmu.edu.cn, sz.mei@nudt.edu.cn,
xinran.li@connect.ust.hk

## Abstract

Generative models have gained significant traction in offline reinforcement learning (RL) due to their ability to model complex trajectory distributions. However, existing generation-based approaches still struggle with long-horizon tasks characterized by sparse rewards. Some hierarchical generation methods have been developed to mitigate this issue by decomposing the original problem into shorter-horizon subproblems using one policy and generating detailed actions with another. While effective, these methods often overlook the multi-scale temporal structure inherent in trajectories, resulting in suboptimal performance. To overcome these limitations, we propose MAGE, a Multi-scale Autoregressive GEneration-based offline RL method. MAGE incorporates a condition-guided multi-scale autoencoder to learn hierarchical trajectory representations, along with a multi-scale transformer that autoregressively generates trajectory representations from coarse to fine temporal scales. MAGE effectively captures temporal dependencies of trajectories at multiple resolutions. Additionally, a condition-guided decoder is employed to exert precise control over short-term behaviors. Extensive experiments on five offline RL benchmarks against fifteen baseline algorithms show that MAGE successfully integrates multi-scale trajectory modeling with conditional guidance, generating coherent and controllable trajectories in long-horizon sparse-reward settings. The source code is available at https://github.com/xmu-rl-3dv/MAGE.

## 1 Introduction

In offline reinforcement learning (RL) (Lange et al., 2012), agents are trained solely from previously collected datasets without further interaction with environments, which makes it attractive for multiple real-world applications, such as robotics (Gupta et al., 2019; Deng et al., 2025; Shen et al., 2023) and clinical medicine (Lee et al., 2025). However, relying on fixed collected datasets to learn a policy faces several challenges, including distributional shift and overestimation bias (Levine et al., 2020; Kumar et al., 2019).

Existing offline RL methods generally fall into four main categories: (1) Generation-based approaches, which view policy learning as conditional trajectory generation (Janner et al., 2021; Chen et al., 2021; Janner et al., 2022; Wang et al., 2023; Ajay et al., 2023; Li et al., 2025); (2) Regularization-based approaches, which aim to prevent policy deviation by adding constraints relative to the behavior policy (Fujimoto et al., 2019; Kumar et al., 2019; Fujimoto & Gu, 2021); (3) Constraint-based approaches, which assign pessimistic values to out-of-distribution actions to suppress their selection (Yang et al., 2021; Kumar et al., 2020); and (4) Model-based approaches, which utilize

---

[*]Equal contribution
[†]Corresponding author

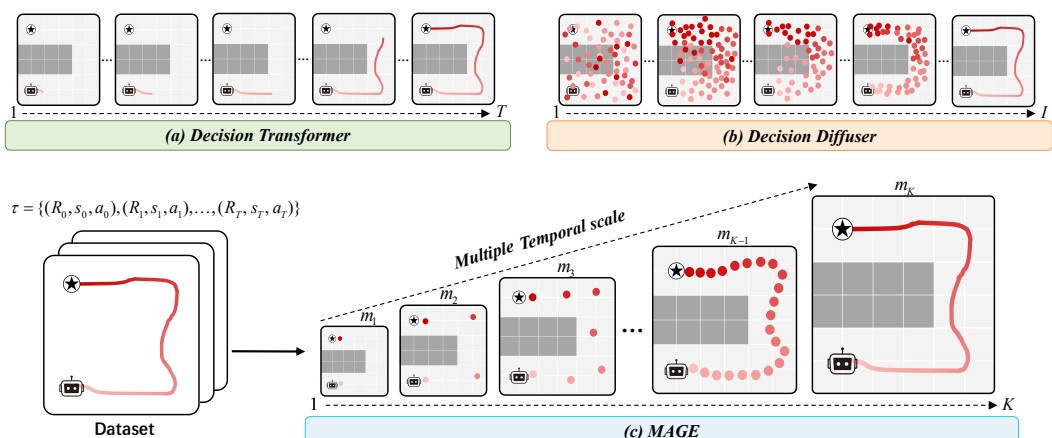

Figure 1: **Schematic Illustration of Generation-based Offline RL Methods.** (a) Decision Transformer follows a step-by-step, autoregressive generation process. (b) Decision Diffuser utilizes an all-at-once, denoising-based generation approach. (c) MAGE operates in a top-down manner, first establishing a macroscopic outline of a trajectory and then progressively refining it with microscopic details.

a learned environment model for planning (Yu et al., 2021; Williams et al., 2016). Our proposed method falls under the generation-based paradigm.

Generation-based offline RL methods, such as Diffusion-QL (Wang et al., 2023), Decision Diffuser (Ajay et al., 2023) and DoF (Li et al., 2025) have shown competitive performance and high trajectory diversity, benefiting from the strong representational capacity of generative models like diffusion processes (Ho et al., 2020; Song et al., 2021) to capture multi-modal distributions. Despite these advantages, such methods struggle in long-horizon sparse-reward tasks, which are prevalent in real-world applications such as robotic manipulation and strategic planning. In such settings, delayed feedback and complex temporal dependencies pose significant challenges for reliable policy learning (Tang et al., 2022; Villaflor et al., 2022; Andersen et al., 2018).

The inferior performance of generation-based offline RL in long-horizon tasks stems primarily from inadequate modeling of multi-scale temporal dependencies, particularly long-range information. While transformers (Chen et al., 2021; Janner et al., 2021) are widely used, their unidirectional autoregressive nature limits bidirectional understanding of the global context. Diffusion models (Wang et al., 2023; Ajay et al., 2023), although generally achieving stronger results, exhibit a local generation bias (Lu et al., 2025), often producing trajectories that are locally plausible but lack global coherence over extended horizons.

A promising direction is to use hierarchical generation methods (HGM), which convert long-horizon tasks into shorter-horizon subproblems. Existing approaches (Ma et al., 2024; Li et al., 2023; Chen et al., 2024) typically adopt a fixed two-layer hierarchy, with each level governed by a distinct policy. For example, HDMI (Li et al., 2023) generates sub-goals at a high level and detailed trajectories at a low level. This rigid structure not only limits the ability to capture multi-scale temporal abstractions, but also introduces significant optimization challenges. The need to jointly optimize two interdependent policies within a fixed hierarchy could lead to training efficiency issues.

To address the challenges of long-horizon and sparse-reward tasks, we propose MAGE, a novel Multi-scale Auto-regressive Generation model for offline RL. MAGE generates trajectories in a top-down coarse-to-fine manner. It produces a long-term, coarse-grained trajectory representation, and then this initial sketch is progressively refined through iterative rounds of auto-regressive generation, with each step yielding a finer-grained representation. Finally, actions are determined based on the resulting multi-scale trajectory representations. Figure 1 provides a schematic overview of this generative process.

MAGE comprises two core components: a multi-scale autoencoder and a multi-scale transformer. The autoencoder encodes a trajectory into a hierarchy of latent representations according to a predefined scale schedule, constituting a set of token maps from coarse-to-fine temporal resolutions. Coarse-

scale tokens capture long-term dependencies, while fine-scale tokens encapsulate short-term details. The multi-scale transformer autoregressively generates these token maps sequentially, with each finer-scale token map conditioned coarser-scale token maps generated in the previous step. This coarse-to-fine generation scheme enables the model to capture both the global trajectory structure and local temporal dynamics, resulting in highly coherent trajectories. For finer-grained control, a condition-guided adapter module is integrated into the decoder, modulating internal representations based on specified conditions to precisely steer the generated trajectories.

Extensive evaluations on five offline RL benchmarks show that MAGE achieves state-of-the-art performance, particularly in long-horizon tasks with sparse rewards, while remaining competitive in dense-reward settings. Systematic ablations confirm the critical role of multi-scale temporal modeling and conditional guidance. Additionally, MAGE exhibits fast inference speeds, providing an efficient and practical solution for complex sequential decision-making.

## 2 BACKGROUND

### 2.1 AUTO-REGRESSIVE MODELS

As a predominant autoregressive architecture, the Transformer (Vaswani et al., 2017) generates sequences by predicting each token $x_i$ solely based on its predecessors $x_{<i}$. The probability of a sequence under this model is defined by:

$$p(x_1, x_2, \ldots, x_T) = \prod_{i=1}^{T} p(x_i | x_1, x_2, \ldots, x_{i-1}) \tag{1}$$

The Visual Autoregressive (VAR) model (Tian et al., 2024) introduces a hierarchical approach to autoregressive data generation. Central to VAR is a hierarchical autoregressive likelihood, which operates across multiple spatial scales rather than individual tokens. The joint probability of generating the complete set of token maps $B = (b_1, b_2, \ldots, b_K)$ is formulated as:

$$p(b_1, b_2, \ldots, b_K) = \prod_{k=1}^{K} p(b_k | b_1, b_2, \ldots, b_{k-1}) \tag{2}$$

Here, each $b_k$ denotes a token map of dimensions $h_k \times w_k$, and the sequence $B$ is ordered according to increasing spatial resolution (i.e., $h_{k+1} > h_k, w_{k+1} > w_k$). The generation of $b_k$ is conditioned on all previously generated coarser-scale maps $b_{<k}$.

### 2.2 VECTOR QUANTIZED VARIATIONAL AUTOENCODER

The Vector Quantized Variational Autoencoder (VQ-VAE) (Van Den Oord et al., 2017) extends the standard Variational Autoencoder (VAE) (Kingma & Welling, 2014) by incorporating tokens (discrete latent representations) through vector quantization. The model comprises three key components: an encoder $E_\phi$ that maps an input $x \in \mathcal{X}$ to a continuous latent vector $z_e = E_\phi(x)$, a learnable codebook $\{e_k\}_{k=1}^{K}$ containing $K$ embedding vectors in $\mathbb{R}^D$, and a decoder $D_\theta$ that reconstructs the input from the quantized codes $\hat{x} = D_\theta(z_q)$. The continuous output $z_e$ is discretized by replacing it with the nearest codebook entry:

$$z_q = \text{Quantize}(z_e) = e_k, \quad k = \arg\min_j \|z_e - e_j\|_2. \tag{3}$$

This quantization step enables the learning of tokens, which are well suited for autoregressive modeling.

## 3 MAGE: MULTI-SCALE AUTO-REGRESSIVE DECISION MAKING

Generation-based offline RL models have demonstrated a competitive advantage in decision-making tasks with complex trajectory distributions. However, they struggle with long-horizon tasks that have sparse rewards. Their lack of long-term awareness and multi-scale modeling for temporal abstraction

often produces trajectories that are locally coherent but globally inconsistent, hindering effective decision-making.

Our key observation is that, for effective long-horizon tasks, it is important to generate trajectories at multiple temporal scales, capturing both long-term and short-term information. We propose MAGE, a **M**ulti-scale **A**utoregressive **GE**neration method for offline RL. MAGE consists of two major modules: multi-scale trajectory autoencoder (Section 3.1) and multi-scale condition-guided auto-regressive generator (Section 3.2).

### 3.1 MULTI-SCALE TRAJECTORY AUTOENCODER

The Multi-scale Trajectory Autoencoder (MTAE) incorporates a multi-scale quantization architecture to capture hierarchical dependencies in long-horizon trajectories. It represents a trajectory $\tau$ as a sequence of state and return-to-go (RTG) pairs: $\tau = (R_0, s_0), (R_1, s_1), \ldots, (R_T, s_T)$, where $R_i$ denotes the cumulative future reward from timestep $i$ onward, and $s_i$ is the corresponding state. To enable autoregressive modeling, MTAE tokenizes the trajectory into discrete representations. This is achieved through a top-down encoding process that maps $\tau$ into a multi-scale token map $M = (m_1, m_2, \ldots, m_K)$. Each token map $m_k \in [V]^{l_k}$ is a sequence of $l_k$ tokens, where each token $t \in [V]$ is an integer from a vocabulary of size $V$. The token map $m_k$ encapsulates temporal information at the $k$-th scale of the trajectory, with $m_1$ capturing the coarsest, global-level structure and $m_K$ containing the finest-grained details.

The encoding and decoding processes of MTAE are depicted in Algorithm 1 and 2. $\mathcal{E}(\cdot)$, $\mathcal{Q}(\cdot)$, and $\mathcal{D}(\cdot)$ denote the encoder, quantizer, and decoder, respectively. MTAE employs a similar architecture to VQVAE (Van Den Oord et al., 2017). Besides, a shared codebook $\mathcal{C}$ is utilized across all scales to ensure that all tokens have the same size and the same vocabulary. The scale-up and scale-down operators are implemented through linear projection. For trajectory modeling, we empirically find that modeling $(R, s)$ rather than other alternatives leads to the highest performance, as shown in Section 4.4.

---

**Algorithm 1** Multi-scale Encoding

**Require:** $\tau = \{(s_0, R_0), \ldots, (s_T, R_T)\}$;
**Require:** temporal scales $[l_k]_{k=1}^K$; codebook $\mathcal{C}$;
1: $f = \mathcal{E}(\tau, R_0)$, $M = []$;
2: **for** $k = 1, \cdots, K$ **do**
3: $\quad m_k = \mathcal{Q}(\text{Scale\_down}(f, l_k))$;
4: $\quad M = \text{queue\_push}(M, m_k)$;
5: $\quad z_k = \text{Lookup}(\mathcal{C}, m_k)$;
6: $\quad z_k = \text{Scale\_up}(z_1, l_K)$;
7: $\quad f = f - z_k$;
8: **end for**
**Ensure:** multi-scale token maps $M$;

---

**Algorithm 2** Multi-scale Decoding

**Require:** multi-scale token maps $M$;
**Require:** temporal scales $[l_k]_{k=1}^K$; codebook $\mathcal{C}$;
1: **for** $k = 1, \cdots, K$ **do**
2: $\quad m_k = \text{queue\_pop}(M)$;
3: $\quad z_k = \text{Lookup}(\mathcal{C}, m_k)$;
4: $\quad z_k = \text{Scale\_up}(z_k, l_K)$;
5: **end for**
6: $Z = (z_1, \cdots, z_K)$
7: $\hat{\tau} = \mathcal{D}(Z, R_0)$;
**Ensure:** reconstructed trajectory $\hat{\tau}$;

---

### 3.2 MULTI-SCALE CONDITIONAL GUIDE AUTOREGRESSIVE GENERATION

MAGE uses multi-scale temporal information as guidance to generate tokens. The token map $m_k$ is generated based on all previous token maps $m_i$ $i < k$ and $(s_0, R_0)$. After generating the token maps, an action $a$ is determined, which is executed by the agent.

In MAGE, a multi-scale conditional guide transformer is tasked with autoregressively predicting the sequence of codebook maps $(m_1, m_2, \ldots, m_K)$. The generative process is described as follows.

$$p(m_1, m_2 \ldots, m_k \mid s_0, R_0) = \prod_{k=1}^{K} p(m_k \mid m_{<k}, s_0, R_0). \tag{4}$$

At each scale $k$, the input to the transformer consists of $s_0$, $R_0$, and the token maps from the previous scale $m_{<k}$. This hierarchical conditioning approach makes the predicted token map $m_k$ close to $R_0$

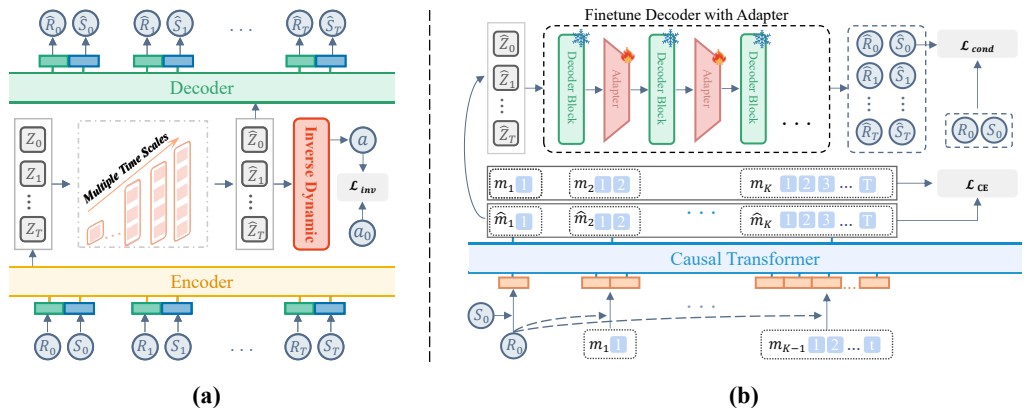

Figure 2: **MAGE Overview:** (a) Multi-scale Representation: hierarchical quantization of trajectories across scales for global–local structure modeling. (b) Condition-guided Decision-making: autoregressive latent prediction with conditional refinement for consistent trajectory generation.

and $s_0$ across different temporal scales. The transformer outputs a categorical distribution over the token map at each scale. It is trained using a cross-entropy loss against the ground-truth token maps.

$$\mathcal{L}_{\text{CE}} = -\sum_{k=1}^{K} \sum_{i=1}^{p_k} \mathbf{m}_{k,i}^{\top} \log \hat{\mathbf{m}}_{k,i}, \tag{5}$$

where $\mathbf{m}_{k,i}$ denotes the one-hot encoding of the ground-truth integer at position $i$ and scale $k$ and $\hat{\mathbf{m}}_{k,i}$ is the predicted categorical distribution. The latent representation $Z = (z_1, \cdots, z_K)$ for the generate trajectory $\hat{\tau}$ is obtained alongside $(m_1, \cdots, m_K)$ through look up in codebook $\mathcal{C}$.

**Determining Action** MAGE adopts a latent inverse dynamics model $I$ to determine the action $a$ to be executed from the generated trajectory. Given the aggregated latent representation $Z$ encoding the generated trajectory, $I$ determines the action as

$$a = I(\sum_{k=1}^{K} z_k), \quad \mathcal{L}_{\text{inv}} = \|a - a_0\|_2^2, \tag{6}$$

$a_0$ is the action taken in $\tau$ at timestep 0. The objective $\mathcal{L}_{\text{inv}}$ is designed to encourage the latent variable $Z$ to preserve dynamics-consistent information at the finest temporal scale for the most recent timestep. As evidenced by the ablation study in Appendix B.5, utilizing this latent representation $Z$ leads to superior performance compared to the use of the fully generated trajectory.

**Condition-Guided Refinement** MAGE generates trajectories starting from the current state $s_0$. However, we identified a challenge: the cross-entropy loss $\mathcal{L}_{CE}$ alone does not guarantee that the first state of the generated trajectory $\hat{\tau}$ exactly matches $s_0$, potentially leading to trajectories that diverge from the intended condition. This issue is compounded by the information loss inherent in the quantization of latent variables $\hat{Z}$. To correct for these deviations, MAGE incorporates an additional condition-guided refinement loss, implemented as a mean squared error between the decoded initial state-return pair and the true initial condition $(s_0, R_0)$.

$$\mathcal{L}_{\text{cond}} = \|\mathcal{D}'(Z, R_0)_0 - (s_0, R_0)\|_2^2. \tag{7}$$

$\mathcal{D}'$ is an augmented decoder with a parameter-efficient refinement module. This decoder maps the latent codes $Z$ back to trajectory space, where $\mathcal{D}'(Z, R_0)_0$ denotes the generated initial condition $(\hat{s}_0, \hat{R}_0)$. To ensure this output strictly matches the true initial condition $(s_0, R_0)$, the conditional loss $\mathcal{L}_{\text{cond}}$ is applied during training. This loss term guides the auto-regressive process to yield conditionally coherent trajectories. The necessity of $\mathcal{L}_{\text{cond}}$ is validated in Appendix Figure 4, where its removal leads to a deviation at the trajectory outset.

## 4 EVALUATION

Comprehensive evaluations against 15 baselines across 5 benchmarks demonstrate MAGE's strong performance in long-horizon sparse-reward tasks, along with competitiveness in dense-reward settings.

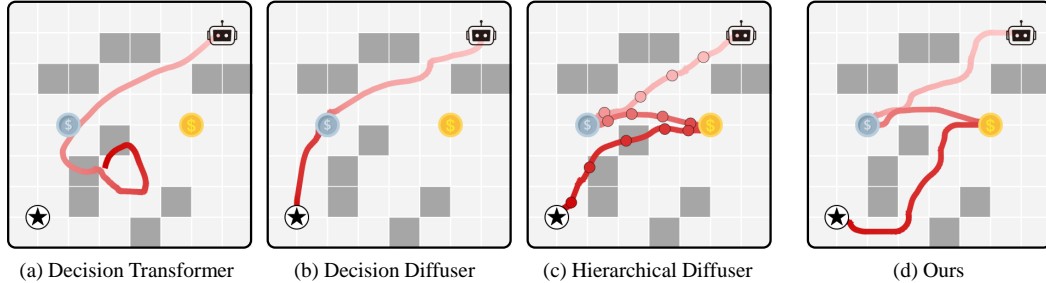

(a) Decision Transformer  (b) Decision Diffuser  (c) Hierarchical Diffuser  (d) Ours

Figure 3: **A Maze Game:** The dark grid cells are walls. The trajectories are plotted in red. Lighter color represents earlier timesteps. The red circles shown in (c) are subgoals.

Ablations validate the necessity of multi-scale modeling and compare trajectory schemes, while results confirm high inference efficiency. Due to space limitations, please refer to Appendix B for details.

## 4.1 EXPERIMENTAL SETUP

**Environment** We evaluate MAGE on widely-used benchmarks covering long-horizon tasks—such as dexterous manipulation (Adroit (Rajeswaran et al., 2018)), sequential tasks (Franka Kitchen (Gupta et al., 2019)), and navigation (Maze2D, Multi2D, AntMaze (Fu et al., 2020))—as well as locomotion tasks with dense rewards (MuJoCo (Todorov et al., 2012)).

**Baselines.** Our method is compared with a broad set of 15 representative baselines covering different families of offline RL approaches.

- **Non-generation methods**: Behavior Cloning(BC) (Bain & Sammut, 1995), Conservative Q-Learning(CQL) (Kumar et al., 2020), and Implicit Q-learning(IQL) (Kostrikov et al., 2022) learn policies directly from offline data via behavior cloning or policy regularization, while Model Predictive Path Integral(MPPI) (Williams et al., 2016) uses environment dynamics for model-based trajectory optimization.

- **Generation-based methods**: Decision Transformer(DT) (Chen et al., 2021), Trajectory Transformer(TT) (Janner et al., 2021) and TAP (Zhang et al., 2023) generate trajectories using Transformers. Diffuser (Janner et al., 2022), Decision Diffuser(DD) (Ajay et al., 2023), and RGG (Lee et al., 2023) generate trajectories using diffusion, while Diffusion-QL (Wang et al., 2023) generates actions through diffusion.

- **Hierarchical generation methods**: ADT (Ma et al., 2024) is a two-level transformer-based method. HDMI (Li et al., 2023) and HD (Chen et al., 2024) are two-level diffusion-based methods. CARP (Gong et al., 2025) is a coarse-to-fine autoregressive modeling method that generates action sequences.

To ensure a fair and meaningful comparison, for each algorithm, we report its performance for each environment reported in the official paper. If such results are unavailable, we report the performance by running the algorithm for some environments.

For all experiments, the average score (with standard error) is utilized to measure the performance of all algorithms. For each environment, results are averaged over 5 random training seeds. In Adroit and Kitchen, each seed is tested 20 times, while in Maze2D, Multi2D, and AntMaze, each seed is tested 100 times due to the stochasticity of the environment.

## 4.2 AN ILLUSTRATIVE EXAMPLE

Figure 3 illustrates a maze navigation task in which the agent must first collect the silver coin, then the gold coin, and finally reach the goal. The agent receives rewards for collecting coins and reaching the goal. The results indicate that Decision Transformer (DT) fails to reach the goal, Decision Diffuser (DD) reaches the goal but fails to discover the gold coin, and Hierarchical Diffuser (HD) produces trajectories that cross walls. MAGE can obtain all the coins and reach the goal. This example

Table 1: The Average Scores for the Adroit Scenarios.

| Scenario | | IQL | DT | ADT | CARP | DD | D-QL | HDMI | HD | MAGE |
|---|---|---|---|---|---|---|---|---|---|---|
| Pen | Expert | 128.0±9.2 | 116.3±1.2 | 113.3±12.1 | 112.7±19.8 | 107.6±7.6 | 112.6±8.1 | 109.5±8.0 | 121.4±14.3 | **147.8±4.9** |
| | Human | 78.4±8.2 | 67.6±5.4 | 70.1±16.1 | 62.3±21.4 | 64.1±9.0 | 66.0±8.3 | 66.2±8.8 | 47.6±14.9 | **137.1±9.0** |
| | Cloned | 83.4±8.1 | 64.4±1.4 | 35.9±13.1 | 12.5±15.2 | 47.7±9.2 | 49.3±8.0 | 48.3±8.9 | 13.9±9.7 | **108.4±17.6** |
| Door | Expert | 106.6±0.3 | 104.8±0.3 | 105.1±0.1 | 98.4±4.7 | 87.0±0.8 | 93.7±0.8 | 85.9±0.9 | 105.9±0.6 | **106.8±0.1** |
| | Human | 3.2±1.8 | 4.4±0.8 | 7.5±2.3 | 5.0±4.6 | 6.9±1.2 | 8.0±1.2 | 7.1±1.1 | 0.2±0.0 | **16.5±0.9** |
| | Cloned | 3.0±1.7 | 7.6±3.2 | 1.8±1.3 | 0.0±0.0 | 9.0±1.6 | 10.6±1.7 | 9.3±1.6 | 4.5±3.6 | **20.5±2.5** |
| Hammer | Expert | 128.6±0.3 | 117.4±6.6 | 127.4±0.4 | 127.5±0.6 | 106.7±1.8 | 114.8±1.7 | 111.8±1.7 | 126.8±1.1 | **131.7±0.2** |
| | Human | 1.7±0.8 | 1.2±0.1 | 1.8±0.2 | 0.9±0.3 | 1.0±0.1 | 1.3±0.1 | 1.2±0.1 | 0.9±0.3 | **10.4±1.2** |
| | Cloned | 1.5±0.6 | 1.8±0.5 | 2.1±0.5 | 0.9±0.2 | 0.9±0.1 | 1.1±0.1 | 1.0±0.1 | 0.9±0.2 | **13.2±4.7** |
| Relocate | Expert | 106.1±4.0 | 104.2±0.4 | 106.4±1.4 | 71.0±8.3 | 87.5±2.8 | 95.2±2.8 | 91.3±2.6 | 62.1±13.1 | **109.6±1.6** |
| | Human | 0.1±0.0 | 0.1±0.0 | 0.1±0.1 | 0.0±0.0 | 0.2±0.1 | 0.2±0.1 | 0.1±0.1 | 0.0±0.0 | **0.3±0.1** |
| | Cloned | **0.0±0.0** | **0.0±0.0** | **0.0±0.0** | -0.2±0.0 | -0.2±0.0 | -0.2±0.0 | -0.1±0.0 | -0.2±0.0 | **0.0±0.0** |
| **Mean(w/o Expert)** | | 21.4 | 18.4 | 14.9 | 10.2 | 16.2 | 17.0 | 16.6 | 8.5 | **38.3** |
| **Mean(all settings)** | | 53.4 | 49.2 | 47.6 | 40.9 | 43.2 | 46.1 | 44.3 | 40.3 | **66.9** |

Table 2: The Average Scores for the Franka Kitchen Scenarios.

| Scenario | | IQL | DT | ADT | CARP | DD | DQL | HDMI | HD | MAGE |
|---|---|---|---|---|---|---|---|---|---|---|
| Kitchen | Partial | 59.7±8.3 | 31.4±19.5 | 64.2±5.1 | 32.5±2.6 | 65.0±2.8 | 60.5±6.9 | - | 73.3±1.4 | **91.3±3.2** |
| | Mixed | 53.2±1.6 | 25.8±5.0 | 69.2±3.3 | 30.0±2.2 | 57.0±2.5 | 62.6±5.1 | 69.2±1.8 | 71.7±2.5 | **86.3±3.3** |
| **Average** | | 56.5 | 28.6 | 66.7 | 31.3 | 61.0 | 61.6 | - | 72.5 | **88.8** |

qualitatively demonstrates the ability of MAGE in such a long-horizon task with sparse rewards. Please refer to Appendix B.4 for further details.

## 4.3 COMPARISON STUDY

We evaluate MAGE against 15 offline RL algorithms on 5 different sets of environments. The results demonstrate MAGE's compelling ability in long-horizon tasks with sparse rewards. Moreover, MAGE is also competitive in tasks with dense rewards.

### 4.3.1 ADROIT: DEXTEROUS MANIPULATION ENVIRONMENTS

The key difficulty of the Adroit environment (Rajeswaran et al., 2018) lies in its sparse reward signals and the requirement for long-horizon, high-dimensional, fine-grained control. As shown in Table 1, MAGE achieves significant improvements on the Pen, Door, and Hammer tasks, with particularly strong performance on Pen, where it substantially outperforms other methods. The results demonstrate that MAGE maintains consistent advantages in Adroit, despite the challenges of sparse rewards and high-dimensional control.

IQL shows limited performance, as it learns value functions, making it suffer from the deadly triad issues (Sutton & Barto, 2018). While DD improves performance through modeling trajectories, its single-step, non-holistic process lacks a global perspective, leading to poor performance. CARP does not model the relationship between generated action sequences and rewards, leading to weaker performance than MAGE. Hierarchical RL methods, such as ADT and HD, do not fully model the multi-scale temporal information in trajectories, leading to suboptimal performance.

MAGE employs trajectory modeling with multi-scale temporal guidance and RTG for high-reward trajectory generation. MAGE achieves the best performance for such long-horizon, high-dimensional, fine-grained control and sparse reward tasks.

Table 3: The Average Scores for the Antmaze Scenarios.

| Scenario | | BC | CQL | IQL | DT | ADT | DD | D-QL | HD | MAGE |
|---|---|---|---|---|---|---|---|---|---|---|
| | U-maze | 47.2±4.0 | 37.2±3.7 | 70.6±3.7 | 51.7±0.4 | 83.0±3.1 | 49.2±3.1 | 66.2±8.6 | 94.0±4.9 | **95.2±2.2** |
| Diverse | Medium | 0.8±0.8 | 67.2±3.5 | 61.7±6.1 | 0.0±0.0 | 83.4±1.9 | 4.0±2.8 | 78.6±10.3 | 88.7±8.1 | **98.2±1.3** |
| | Large | 0.0±0.0 | 20.5±13.2 | 27.6±7.8 | 0.0±0.0 | 65.4±4.9 | 0.0±0.0 | 56.6±7.6 | 83.6±5.8 | **84.6±3.6** |
| | U-maze | 55.2±4.1 | 92.7±1.9 | 83.3±4.5 | 57.0±9.8 | 83.8±2.3 | 73.1±2.5 | 93.4±3.4 | 72.2±2.0 | 92.2±2.7 |
| Play | Medium | 0.0±0.0 | 65.7±11.6 | 64.6±4.9 | 0.0±0.0 | 82.0±1.7 | 8.0±4.3 | 76.6±10.8 | 42.0±1.9 | **92.0±2.7** |
| | Large | 0.0±0.0 | 20.7±7.2 | 42.5±6.5 | 0.0±0.0 | 71.0±1.3 | 0.0±0.0 | 46.4±8.3 | 54.7±2.0 | **75.8±4.3** |
| **Average** | | 17.2 | 50.7 | 58.4 | 18.1 | 78.1 | 22.4 | 69.6 | 72.5 | **89.7** |

Table 4: The Average Scores for the Maze2D and Multi2D Scenarios.

| Scenario | | CQL | IQL | DT | ADT | CARP | DD | HDMI | HD | MAGE |
|---|---|---|---|---|---|---|---|---|---|---|
| | U-maze | -8.9±6.1 | 42.1±0.5 | 31.0±21.3 | 60.5±2.0 | 26.2±3.9 | 116.2±2.7 | 120.1±2.5 | 128.4±3.6 | **145.4±3.2** |
| Maze2D | Medium | 86.1±9.6 | 34.8±2.7 | 8.2±4.4 | 109.4±6.2 | 65.8±2.4 | 122.3±2.1 | 121.8±1.6 | 135.6±3.0 | **155.0±3.3** |
| | Large | 23.7±36.7 | 61.7±3.5 | 2.3±0.9 | 155.4±10.4 | 0.7±2.0 | 125.9±1.6 | 128.6±2.9 | 155.8±2.5 | **159.4±2.9** |
| **Single-task Average** | | 33.6 | 46.2 | 13.8 | 108.4 | 30.9 | 121.5 | 123.5 | 139.9 | **153.3** |
| | U-maze | 25.4±5.8 | 13.5±3.0 | 15.6±2.4 | 66.9±5.2 | 82.5±3.4 | 128.2±2.1 | 131.3±1.8 | 144.1±1.2 | **150.4±1.8** |
| Multi2D | Medium | 8.3±3.9 | 8.3±3.4 | 6.3±1.6 | 108.5±6.2 | 48.7±2.6 | 129.7±2.7 | 131.6±1.9 | 140.2±1.6 | **147.7±3.1** |
| | Large | 8.2±4.2 | 5.2±1.4 | 5.1±1.4 | 159.4±9.2 | 32.2±3.2 | 130.5±4.2 | 135.4±2.5 | 165.5±0.5 | **166.8±3.6** |
| **Multi-task Average** | | 14.0 | 9.0 | 9.0 | 111.6 | 54.5 | 129.5 | 132.8 | 149.9 | **155.0** |

### 4.3.2 FRANKA KITCHEN: COMPOSITIONAL ENVIRONMENTS

In the Franka Kitchen environments (Gupta et al., 2019), success depends not only on reaching individual sub-goals but also on executing them in the correct order, which makes naive trajectory generation prone to errors. By leveraging multi-scale trajectory generation, MAGE captures both the global task structure and local sub-goal details, providing coherent and fine-grained decision-making. MAGE demonstrates superior performance, surpassing all competing algorithms by a considerable margin, as detailed in Table 2.

### 4.3.3 MAZE NAVIGATION ENVIRONMENTS: ANTMAZE, MAZE2D, AND MULTI2D

In the AntMaze, Maze2D, and Multi2D scenarios (Fu et al., 2020), a robot must navigate through maze-like structures to reach a distant goal location. Mazes of different sizes (U-shaped, medium, and large) are evaluated. In the AntMaze tasks, the proposed MAGE outperforms the baselines on 5 out of 6 datasets, as shown in Table 3. MAGE performs the best on all datasets for the Maze2D and Multi2D scenarios, as shown in Table 4. The results demonstrate that our method can effectively handle the long-horizon navigation tasks.

Beyond sparse-reward, long-horizon tasks, MAGE also demonstrates strong performance in environments with dense rewards, as detailed in Appendix B.3 for Gym locomotion tasks. Notably, MAGE achieves top performance in 7 out of 9 tasks, confirming its general competitiveness across different reward structures.

### 4.4 ABLATION STUDY

To explore the effectiveness of design components, we conduct ablation studies regarding temporal scale, generation scheme, and conditional guidance on the Adroit scenarios: Pen-Expert and Door-Cloned. Moreover, we evaluate the inference time of MAGE.

**Importance of multiple temporal scales** $(K)$. We analyze the impact of varying the number of temporal scales $K$. Results in Table 5 show that performance generally improves as $K$ increases up to 8, confirming that modeling multiple temporal scales is beneficial. However, beyond this point (e.g., for Door-Cloned), performance declines. This suggests that while incorporating finer-grained information helps up to a certain point, excessive granularity ($K \geq 8$) may introduce noise or unnecessary complexity without further gains. The optimal $K$ is thus task-dependent.

Table 5: Performance of varying the number of temporal scales $K$

| Scenario | ADT | HD | 1 | 2 | 4 | 6 | 8 | 10 |
|---|---|---|---|---|---|---|---|---|
| Pen-Expert | 113.3±12.1 | 121.4±14.3 | 123.5±9.1 | 127.5±5.2 | 134.2±7.7 | 139.5±5.7 | 147.8±4.9 | 149.9±9.2 |
| Door-Cloned | 1.8±1.3 | 4.5±3.6 | 5.2±1.8 | 6.0±2.1 | 10.7±2.3 | 14.0±2.7 | 20.5±2.5 | 17.0±2.7 |

Table 6: Different trajectory sequence generation schemes.

| Scenario | ADT | HD | S | A | A+CQL | R, S, A | Ours |
|---|---|---|---|---|---|---|---|
| Pen-Expert | 113.3±12.1 | 121.4±14.3 | 127.7±3.3 | 127.1±13.0 | 127.6±4.6 | 124.9±7.8 | 147.8±4.9 |
| Door-Cloned | 1.8±1.3 | 4.5±3.6 | 11.9±2.7 | 9.1±2.3 | 4.9±1.0 | 17.2±3.0 | 20.5±2.5 |

Table 7: Ablation results of condition and constraint loss on the Adroit scenarios

| Scenario | ADT | HD | $\not{R}$ in $\mathcal{D}$ | $\not{R}$ in $m_{k>1}$ | $\not{R}$ in $\mathcal{L}_{cond}$ | Ours |
|---|---|---|---|---|---|---|
| Pen-Expert | 113.3 ± 12.1 | 121.4±14.3 | 140.3±9.1 | 139.5±6.7 | 139.9±7.2 | 147.8±4.9 |
| Door-Cloned | 1.8 ± 1.3 | 4.5±3.6 | 12.3±1.9 | 16.3±2.6 | 17.1±2.5 | 20.5±2.5 |

Table 8: Average inference time (ms) in Adroit environments.

| Method | Ours | DT | TT | ADT | DD | HD |
|---|---|---|---|---|---|---|
| Time (ms) | 27.30±0.69 | 6.49±0.11 | 12863.07±19.37 | 7.81±0.14 | 2339.16±11.37 | 1480.21±25.18 |

**Comparison of Trajectory Sequence Generation Schemes.** We benchmark various sequence modeling strategies to identify the most effective scheme for trajectory generation. As shown in Table 6, these include modeling states only (S), actions only (A), actions with CQL regularization (A+CQL), and the joint modeling of returns, states, and actions (R, S, A). Our approach, which models returns and states (R, S), achieves the best performance. This result suggests that the (R, S) scheme optimally balances the capture of high-level outcome intent (via returns) with detailed environmental dynamics (via states), whereas incorporating actions adds unnecessary complexity that hinders performance.

**Role of RTG-based Conditioning.** To quantify the importance of return-to-go (RTG) guidance, we systematically ablate its use in three key parts of MAGE: the autoencoder ($\not{R}$ in $\mathcal{D}$), the multi-scale transformer for finer scales ($\not{R}$ in $m_{k>1}$), and the conditioning loss $\mathcal{L}_{cond}$ ($\not{R}$ in $\mathcal{L}_{cond}$). The results (Table 7) indicate a consistent performance drop when RTG is removed, underscoring its critical role in aligning the generated trajectories with the desired return across temporal scales.

**Inference Speed.** As summarized in Table 8, MAGE achieves a favorable balance between performance and efficiency. It runs approximately 50× faster than HD and 80× faster than DD. While slightly slower than some other Transformer-based methods, MAGE maintains a low inference time of 27 ms per step. This rate is well within the 20 Hz requirement for real-time robotic control (Reed et al., 2022), demonstrating its practical applicability.

## 5 RELATED WORK

### 5.1 GENERATION-BASED OFFLINE RL

Offline RL (Kostrikov et al., 2022; Kumar et al., 2020; Fujimoto et al., 2018; Fujimoto & Gu, 2021; Fujimoto et al., 2019; Kumar et al., 2019) aims to learn policies from static datasets. A prominent branch of work is generation-based methods (Chen et al., 2021; Lee et al., 2023; Ye & Gombolay, 2024; Zhang et al., 2023), which leverage generative models like Transformers (Vaswani et al., 2017), flows (Kingma & Dhariwal, 2018), and diffusion models (Ho et al., 2020) to model the data distribution. Among these, diffusion-based approaches have been widely adopted.

Despite their powerful modeling capacity, diffusion-based RL methods face notable challenges. Approaches like Diffusion-QL (Wang et al., 2023) learn a policy with Q regularization, while Diffuser (Janner et al., 2022), Decision Diffuser (Ajay et al., 2023), and RGG (Lee et al., 2023) generate trajectories for planning. However, they are plagued by a local generation bias (Lu et al.,

2025), which can compromise global coherence, especially in long-horizon sparse-reward tasks. Their iterative denoising also results in slow inference.

## 5.2 HIERARCHICAL RL

Recent hierarchical methods in offline RL are often inspired by human decision-making processes. Hierarchical offline RL (Ajay et al., 2021; Rao et al., 2022) decomposes long-horizon tasks into manageable subproblems, which can be broadly categorized as subgoal-based or skill-based (Hutsebaut-Buysse et al., 2022). Subgoal-based methods identify intermediate targets (Pateria et al., 2020), while skill-based approaches learn reusable low-level behaviors (Villecroze et al., 2022). Although MAGE can be viewed as subgoal-based, it differs fundamentally by learning a single unified policy across all latent temporal hierarchies, rather than separate policies for each level.

Existing two-level hierarchical models include HDT (Correia & Alexandre, 2023) and ADT (Ma et al., 2024), which use an autoregressive framework: a high-level policy generates subgoals or prompts, and a low-level policy produces actions conditioned on them. Similarly, HDMI (Li et al., 2023) and HD (Chen et al., 2024) employ a diffusion-based two-stage process, first generating subgoals under reward guidance and then producing subgoal-conditioned trajectories. While effective, these methods often fail to capture the full spectrum of multi-scale temporal dependencies in long-horizon tasks.

CARP (Gong et al., 2025) is a multi-level method that generates action sequences based on current state. Due to their high-frequency and non-smooth characteristics, action sequences (like joint torques) are considerably more difficult to predict (Ajay et al., 2023; Tedrake, 2009). Moreover, without explicit return conditioning, the approach cannot guarantee high returns. In contrast, MAGE utilizes a return-conditioned, multi-scale auto-regressive process over states and RTG, ensuring high-performance outcomes. Please refer to Appendix B.7 for more in-depth discussion of MAGE and others.

## 6 DISCUSSION

MAGE provides a coarse-to-fine framework for long-horizon trajectory modeling, achieving strong performance across diverse benchmarks. However, its hierarchical design involves natural trade-offs, as committing to global plans at coarse scales can limit the flexibility of fine-scale refinements. In environments with extremely sparse rewards and long horizons (i.e., OGBench (Park et al., 2025)). We compared several strong algorithms on these difficult tasks and found that MAGE performs competitively (Table 24), although handling such extreme scenarios remains an open challenge. Moreover, managing distribution shifts in out-of-distribution scenarios remains a critical challenge, and further research is needed to better handle such situations.

The multi-scale mechanism is extensible to multi-agent reinforcement learning research (Qin et al., 2024; Shen et al., 2022). The MAGE framework provides a flexible, effective way for capturing intricate multi-agent coordination patterns (Qin et al., 2025).

## 7 CONCLUSION

We propose MAGE, a multi-scale autoregressive generation method for offline reinforcement learning. It consists of a multi-scale condition-guide autoencoder and a multi-scale transformer. The transformer generates trajectories in a multi-time-scale approach conditioning on return-to-goal and current state. Extensive experiments on five offline RL benchmarks against fifteen approaches validate the effectiveness of MAGE. The results demonstrate that MAGE successfully integrates multi-scale trajectory modeling with conditional guidance, enabling the generation of coherent and controllable trajectories, and could effectively handle tasks with long horizons and sparse rewards.

## ACKNOWLEDGMENTS

This work was partially supported by the Fundamental Research Funds for the Central Universities (No. 20720230033), by Xiaomi Young Talents Program. We would like to thank the anonymous reviewers for their valuable suggestions.

REPRODUCIBILITY STATEMENT

Pseudocode and framework diagrams of our proposed method are provided in Appendix A, allowing readers to understand the algorithmic structure and workflow. All datasets used in our experiments are publicly available from the D4RL (Fu et al., 2020) benchmark. Detailed hyperparameter settings for training and evaluation can be found in Appendix B. We have included the source code of MAGE in the supplementary.

ETHICS STATEMENT

This work focuses on developing and evaluating reinforcement learning methods in simulated environments. Our study does not involve human subjects, personally identifiable information, or sensitive data. The datasets and benchmarks used are publicly available and widely adopted in the reinforcement learning community. We believe that our research does not raise ethical concerns related to privacy, fairness, or potential misuse.

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

# Appendix

## A ALGORITHM

For completeness, we provide detailed algorithmic descriptions and the overall framework in this section. Together, these pseudocode listings provide a transparent view of both the quantization and latent prediction stages, complementing the high-level descriptions in the main text. They serve as a step-by-step reference for reproducing our method and clarifying the implementation details that underpin the proposed framework.

In addition, the framework illustration (Figure 2) offers an intuitive overview of how these components interact, highlighting the multi-scale representation and condition-guided generation process that form the core of MAGE.

### A.1 BACKGROUND

#### A.1.1 VECTOR QUANTIZED VARIATIONAL AUTOENCODER

The Vector Quantized Variational Autoencoder (VQ-VAE) (Van Den Oord et al., 2017) encodes an input into a discrete tokens. It extends the standard Variational Autoencoder (VAE) (Kingma & Welling, 2014) by introducing discrete latent variables through vector quantization. It consists of an encoder $E_\phi$ that maps an input $x \in \mathcal{X}$ to a continuous latent $z_e = E_\phi(x) \in \mathbb{R}^D$, a learnable codebook with $K$ embedding vectors $\{e_k\}_{k=1}^K$ where $e_k \in \mathbb{R}^D$, and a decoder $D_\theta$ that maps discrete codes $z_q$ to a reconstruction $\hat{x} = D_\theta(z_q)$. Each encoder output $z_e$ is replaced by its nearest neighbor in the codebook to obtain the discrete token $z_q$:

$$z_q = \text{Quantize}(z_e) = e_k, \quad k = \arg\min_j \|z_e - e_j\|_2. \tag{A.1}$$

This enables the model to learn discrete representations suitable for autoregressive modeling.

The VQ-VAE is trained by jointly optimizing reconstruction quality and aligning encoder outputs with their assigned codebook vectors. The overall loss is

$$\mathcal{L} = \|x - \hat{x}\|_2^2 + \|\text{sg}[z_e] - e_k\|_2^2 + \beta\|z_e - \text{sg}[e_k]\|_2^2, \tag{A.2}$$

where $\text{sg}[\cdot]$ denotes the stop-gradient operator and $\beta$ controls the encoder's commitment to codebook entries. Gradients are copied from $z_q$ to $z_e$ during backpropagation, enabling end-to-end learning despite the discrete bottleneck.

### A.2 MAGE FRAMEWORK

The MAGE(Multi-scale Autoregressive GEneration) framework is designed to model trajectories in a hierarchical and multi-scale manner, capturing both global structures and local dynamics.

As shown in Figure 2(a), trajectories are first quantized across multiple scales, where higher-level latents encode coarse, long-horizon patterns, while lower-level latents capture fine-grained, short-horizon variations. This hierarchical representation enables the model to preserve coherent structures across temporal scales while effectively propagating high-level information to guide the generation of long-horizon trajectories.

Figure 2(b) illustrates the conditional trajectory generation process in MAGE. At each scale, MAGE autoregressively predicts latent variables conditioned on past latents and return-to-go. These predictions are then refined and fine-tuned to better align the generated trajectories with the provided conditions. The refined latents are finally decoded into states, producing coherent trajectories across scales.

### A.3 MULTI-SCALE QUANTIZATION

The first part presents the multi-scale quantization procedure of our VQ-VAE framework.

This pseudocode illustrates how the encoder maps a trajectory into hierarchical continuous embeddings, how these embeddings are quantized across scales using codebook lookups, and how the

discrete hierarchy is aggregated into the final latent representation $\hat{f}$ for trajectory reconstruction. The pseudocode emphasizes the residual connections across scales, ensuring that finer levels progressively refine the coarser representations while mitigating information loss.

---

**Algorithm 3** Multi-scale Encoding

---

**Require:** raw Trajectory $\tau = \{(s_0, R_0), (s_1, R_1), \ldots, (s_T, R_T)\}$; Number of scales $K$, scales $l_{k\,k=1}^{K}$; codebook $\mathcal{C}$
1: $f = \mathcal{E}(\tau, R_0)$, $M = []$, $Z = []$;
2: **for** $k = 1, \cdots, K$ **do**
3: $\quad m_k = \mathcal{Q}(\text{Scale\_down}(f, l_k))$;
4: $\quad M = \text{queue\_push}(M, m_k)$;
5: $\quad z_k = \text{Lookup}(\mathcal{C}, m_k)$;
6: $\quad z_k = \text{Scale\_up}(\hat{z_k}, l_K)$;
7: $\quad f = f - z_k$;
8: **end for**
**Ensure:** multi-scale token maps $M$;

---

**Algorithm 4** Multi-scale Decoding

---

**Require:** multi-scale token maps $M$; Number of scales $K$, scales $l_{k\,k=1}^{K}$; current state $s_0$; codebook $\mathcal{C}$
1: **for** $k = 1, \cdots, K$ **do**
2: $\quad m_k = \text{queue\_pop}(M)$;
3: $\quad z_k = \text{Lookup}(\mathcal{C}, m_k)$;
4: $\quad z_k = \text{Scale\_up}(z_k, l_K)$;
5: **end for**
6: $Z = (z_1, \cdots, z_K)$
7: $\hat{\tau} = \mathcal{D}(Z, R_0)$;
**Ensure:** reconstructed trajectory $\hat{\tau}$;

---

## A.4 AUTOREGRESSIVE GENERATION PROCESS

The second part of the appendix provides the pseudocode for the Transformer-based latent prediction process. In this procedure, the Transformer autoregressively predicts codebook indices conditioned on the initial state $s_0$ and target return $R_0$, thereby capturing temporal dependencies in the discrete latent space.

The pseudocode explicitly demonstrates how predicted indices are embedded into tokens, how the model outputs categorical distributions over codebook entries, and how the training is performed with cross-entropy loss against the ground-truth indices.

It further outlines the inference phase, where the most probable indices are selected, the corresponding codebook vectors are retrieved, and the resulting multi-scale latent representation is decoded to reconstruct the trajectory.

### A.4.1 GUIDING MODELS WITH CONDITIONAL CONSTRAINTS

While the cross-entropy loss enforces consistency between predicted and ground-truth latent variables, it does not guarantee that the generated trajectory strictly matches the prescribed initial state $s_0$ and target $RTG$. As a result, the generated rollouts may deviate from the desired conditional targets. Moreover, since the latent variables are discrete and quantized representations, information loss is inevitable; even perfectly predicted latents can still lead to biased reconstructions. To address these limitations, we introduce a condition-guided refinement mechanism that enables end-to-end optimization, ensuring that generated trajectories adhere both to the autoregressive latent dynamics and to the input conditions.

Concretely, we decode the autoregressively predicted multi-scale latents $Z$ using the decoder $\mathcal{D}$, with its parameters frozen to preserve the trajectory prior it has learned. However, a fixed decoder cannot

---

**Algorithm 5** Training Multi-scale Transformer with Cross-Entropy and Conditional Fine-Tuning

---

**Require:** training trajectories $\{\tau\}$; VQ-VAE encoder for ground-truth tokens; codebook $\mathcal{C}$; number of scales $K$; scales $\{l_k\}_{k=1}^K$; hyperparameters $\lambda_{\text{cond}}$

1: initialize Transformer parameters $\theta$ and adapter parameters $\phi$
2: **for** each mini-batch of trajectories **do**
3:     $\mathcal{L}_{\text{CE}} \leftarrow 0$
4:     **for** each trajectory $\tau$ in the mini-batch **do**
5:         obtain ground-truth multi-scale tokens maps $M^{gt} = \{m_{k,i}^{gt}\}_{k=1,i=1}^{K,l_k}$ using the VQ-VAE encoder
6:         $M_{\text{soft}} \leftarrow []$
7:         **for** $k = 1, \cdots, K$ **do**
8:             **for** $i = 1, \cdots, p_k$ **do**
9:                 prepare input tokens with $s_0, R_0, M_{\text{soft}}$
10:                obtain logits $\mathbf{l}_{k,i} \leftarrow \text{Transformer}_\theta(\text{tokens})$
11:                compute predicted categorical distribution $\hat{\mathbf{m}}_{k,i} = \text{softmax}(\mathbf{l}_{k,i})$
12:                accumulate cross-entropy loss $\mathcal{L}_{\text{CE}} += -\log \hat{\mathbf{m}}_{k,i}[m_{k,i}^{gt}]$
        // Straight-through estimator
13:                $y_{k,i}^{\text{soft}} \leftarrow \text{GumbelSoftmax}(\mathbf{l}_{k,i}, \tau_g)$
14:                $m_{k,i}^{\text{hard}} \leftarrow \arg\max(y_{k,i}^{\text{soft}})$
15:                construct one-hot $y_{k,i}^{\text{hard}}$ with 1 at index $m_{k,i}^{\text{hard}}$
16:                apply STE: $y_{k,i} \leftarrow y_{k,i}^{\text{hard}} - \text{stopgrad}(y_{k,i}^{\text{soft}}) + y_{k,i}^{\text{soft}}$
17:                compute soft codebook vector $z_{k,i} \leftarrow y_{k,i}^\top \mathcal{C}$
18:             **end for**
19:             scale up $z_k \leftarrow \text{Scale\_up}(z_k, l_k)$
20:             append soft token $M_{\text{soft}} \leftarrow \text{queue\_push}(M_{\text{soft}}, y_k)$
21:         **end for**
22:         $Z \leftarrow (z_1, \cdots, z_K)$
23:         decode with adapter-augmented decoder $\hat{\tau} \leftarrow \mathcal{D}'(Z, R_0)$
24:         compute condition loss $\mathcal{L}_{\text{cond}} \leftarrow \|(R_0, s_0) - \hat{\tau}_0\|_2^2$
25:     **end for**
26:     total loss $\mathcal{L} \leftarrow \mathcal{L}_{\text{CE}} + \lambda_{\text{cond}}\mathcal{L}_{\text{cond}}$
27:     update $\theta$ and $\phi$ by descending gradient of $\mathcal{L}$
28: **end for**
**Ensure:** trained Transformer parameters $\theta$ and adapter parameters $\phi$

---

dynamically adapt to specific conditional inputs, limiting its ability in conditional generation. Inspired by parameter-efficient fine-tuning, we insert lightweight adapter (Houlsby et al., 2019) modules between decoder layers. These adapters specialize in modulating internal representations according to the conditional signals, thereby enhancing the sensitivity of the decoder to $s_0$ and the target RTG.

To enable gradient propagation through the inherently non-differentiable codebook lookup, we adopt the Gumbel-Softmax relaxation. Instead of sampling hard indices directly, we draw differentiable approximations from the categorical distribution. In the forward pass, the straight-through estimator discretizes these samples via `argmax` for codebook indexing, while in the backward pass, gradients are computed with respect to the continuous relaxation. This mechanism preserves the discrete structure required for decoding while ensuring differentiability, thereby allowing end-to-end optimization under conditional constraints.

Under this end-to-end differentiable framework, the latents $Z$ is decoded through the adapter-augmented decoder to produce the final trajectory $\hat{\tau}$. To enforce strict conditional alignment, we introduce a condition loss defined as the mean squared error between the decoded initial state-return pair and the target condition.

$$\mathcal{L}_{\text{cond}} = \|\mathcal{D}'(Z)_0 - (s_0, R_0)\|_2^2. \tag{A.3}$$

Here, $\mathcal{D}'$ denotes the decoder equipped with adapters, and $Z$ represents the latent representation retrieved from the codebook based on the indices predicted by the model. This loss not only adapts

the decoder to the current condition but also guides the latent prediction process, encouraging the model to compose optimal discrete tokens from the fixed codebook so that the decoded trajectory precisely satisfies the specified initial conditions.

# B    EXPERIMENT DETAILS

## B.1    EXPERIMENTAL SETUP

We describe the baseline algorithms used in our experiments in detail, organized into three categories based on their approach.

- **Non-generation-based Methods.** This category includes approaches that do not rely on explicit generative modeling. BC learns policies by directly imitating expert demonstrations through supervised learning, while CQL regularizes Q-value estimation to prevent overestimation and improve stability in offline settings. IQL further decouples Q-function learning from policy updates, achieving better robustness by avoiding direct policy constraints. In contrast, MPPI (Williams et al., 2016) is a model-based control method which uses learned or known dynamics to sample candidate trajectories and optimizes them with a cost function, representing a trajectory optimization approach rather than a pure policy learning method.

- **Generation-based methods.** These methods reframe offline RL as a conditional generative modeling task, learning to generate trajectories or actions conditioned on states and task signals (e.g., desired returns or Q-values) instead of directly optimizing a policy. DT and TT employ Transformer architectures to model long-horizon dependencies. DT conditions on desired returns and generates actions autoregressively, while TT focuses on trajectory-level prediction by learning a sequence model over state–action pairs. Diffuser and Decision Diffuser adopt diffusion models to synthesize entire trajectories under task constraints; the latter incorporates classifier-free guidance to balance multiple conditional signals. Diffusion-QL differs by integrating Q-learning signals directly into the diffusion process, generating high-value actions rather than full trajectories, thereby bridging generative modeling with value-based RL.

- **Hierarchical generation methods.** Long-horizon tasks are particularly challenging for flat generative models, motivating hierarchical designs. ADT introduces a hierarchical reinforcement learning framework, where a high-level policy generates prompts that guide a low-level Decision Transformer to produce actions, thus enhancing the ability to stitch suboptimal trajectories into coherent solutions. HDMI leverages graph-based planning to extract subgoals and incorporates them into diffusion-based trajectory generation. HD improves diffusion planning efficiency by using jumpy hierarchical planning to expand the temporal horizon effectively. CARP uses a coarse-to-fine generation strategy for imitation learning, first producing coarse action chunks and then refining them into precise actions.

The work used for comparison is listed as shown in table 9.

### B.1.1    COMPUTING RESOURCES

The experimental work was carried out on a high-performance computing cluster that includes several NVIDIA GeForce RTX 4090 GPUs to supply the required computing power. The cluster is also

---

[1]https://github.com/BY571/CQL

[2]https://github.com/ikostrikov/implicit_q_learning

[3]https://github.com/kzl/decision-transformer

[4]https://github.com/ZhengyaoJiang/latentplan

[5]https://github.com/mamengyiyi/Autotuned-Decision-Transformer

[6]https://github.com/JannerM/trajectory-transformer

[7]https://github.com/jannerm/diffuser

[8]https://github.com/leekwoon/rgg

[9]https://github.com/anuragajay/decision-diffuser

[10]https://github.com/HeyuanMingong/DiffusionQL

[11]https://github.com/changchencc/Simple-Hierarchical-Planning-with-Diffusion

[12]https://github.com/ZhefeiGong/carp

Table 9: Baseline algorithms

| No. | Algorithm | Brief Description |
|---|---|---|
| 1 | BC (Bain & Sammut, 1995) | Trains a model by directly learning from examples provided by an expert, enabling the model to mimic the expert's behavior |
| 2 | CQL[1] (Kumar et al., 2020) | Updates Q-values conservatively to improve stability and sample efficiency |
| 3 | IQL[2] (Kostrikov et al., 2022) | Decouples policy updates from Q-value estimation to improve the stability and performance of offline reinforcement learning |
| 4 | DT[3] (Chen et al., 2021) | Uses a Transformer architecture to model sequences of states, actions, and rewards, enabling it to make decisions based on the entire history of interactions and desired outcomes |
| 5 | TAP[4] (Zhang et al., 2023) | Addresses the challenge of high-dimensional control by planning over temporally abstract latent actions, drastically reducing decision latency while improving performance |
| 6 | ADT[5] (Ma et al., 2024) | Jointly optimizes high-level prompt and low-level action policies to improve Decision Transformer's ability to stitch trajectories |
| 7 | TT[6] (Janner et al., 2021) | Predicts future states and actions by modeling sequences of past trajectories |
| 8 | MPPI (Williams et al., 2016) | Uses a probabilistic approach to optimize control inputs by sampling future trajectories and selecting the best one based on a cost function |
| 9 | Diffuser[7] (Janner et al., 2022) | Leverages diffusion models to generate high-reward trajectories conditioned on past experiences and guided by rewards |
| 10 | RGG[8] (Lee et al., 2023) | Improves diffusion-based planners by training a "gap predictor" that guides trajectory generation away from unreliable plans |
| 11 | Decision Diffuser[9] (Ajay et al., 2023) | Incorporates classifier-free guidance to dynamically fuse multiple conditions, enabling more flexible and diverse policy generation |
| 12 | Diffusion QL[10] (Wang et al., 2023) | Combines behavior cloning with Q-learning guidance during training to generate high-value actions through an iterative denoising process |
| 13 | HDMI (Li et al., 2023) | Extracts subgoals by using a graph-based planning method that constructs a weighted graph from the dataset and finds optimal subgoal sequences through shortest path search |
| 14 | HD[11] (Chen et al., 2024) | Improves diffusion planning efficiency and generalization in long-horizon tasks by using a jumpy planning strategy |
| 15 | CARP[12] (Gong et al., 2025) | Introduces a coarse-to-fine generation method for the refinement of action chunks |

equipped with Intel(R) Xeon(R) Gold 6348 CPUs, each operating at a frequency of 2.60GHz. In order to verify the reliability of our findings, we executed our algorithm on five separate occasions for each experimental configuration, utilizing distinct random seeds each time.

### B.1.2 ENVIRONMENT AND HYPERPARAMETERS

Our method is implemented based on the source code of DT (Chen et al., 2021), TAP (Zhang et al., 2023) and VAR (Tian et al., 2024).

The training hyperparameters for the Adroit environment in trajectory generation are shown in Table 10. The training hyperparameters for the Kitchen environment in trajectory generation are shown in Table 11. The training hyperparameters for the Antmaze environment in trajectory generation are shown in Table 12. The training hyperparameters for the Maze2D and Multi2D environment in trajectory generation are shown in Table 13.

The learning rate controls the step size during optimization, ensuring stable updates to model parameters. The horizon specifies the number of future steps considered in trajectory prediction, which is crucial in tasks requiring long-term reasoning. Dropout is used to prevent overfitting and improve generalization, while the discount factor balances immediate and long-term rewards. Higher

values encourage the agent to prioritize delayed outcomes, which are common in sparse reward scenarios. The codebook size defines the capacity of discrete latent representations for trajectory modeling, and the number of Transformer blocks, attention heads, and embedding dimensions together determine the expressive power of the sequence model. Batch size influences both training stability and computational efficiency, while the Adam optimizer is employed to adaptively adjust learning rates during training. All experiments use datasets provided by the D4RL benchmark suite (Fu et al., 2020), which standardizes offline reinforcement learning evaluation and ensures comparability across methods.

**Adroit** The Adroit benchmark focuses on dexterous robotic hand manipulation and is widely regarded as one of the most challenging domains in offline reinforcement learning due to its high-dimensional continuous action space, contact-rich dynamics, and extremely sparse rewards. It contains four primary tasks that evaluate different aspects of fine-grained control. In the *pen* task, the robot hand must rotate and manipulate a pen to match a desired orientation, requiring precise finger coordination and continuous adjustment of forces. The *door* task involves opening a door by turning the handle and pulling it open, which demands both grasping and forceful motion under complex dynamics. The *hammer* task requires the agent to pick up a hammer and strike a nail into a board, posing difficulties due to unstable contact interactions and the need for accurate motion sequencing. Finally, the *relocate* task challenges the agent to grasp a ball and place it at a designated target location, which requires the integration of grasping, lifting, and accurate placement in three-dimensional space. Together, these tasks test whether an algorithm can generate coherent long-horizon action sequences that achieve realistic manipulation skills.

Each task is available with three types of datasets that reflect different data collection strategies. The *expert* dataset is collected from demonstrations generated by a near-optimal policy and represents high-quality trajectories that closely follow the desired behavior. The *human* dataset is collected from human teleoperation, resulting in diverse but suboptimal demonstrations that include natural variations and mistakes. The *cloned* dataset is generated by a behavior cloning policy trained on human demonstrations, which often produces noisy and inconsistent trajectories due to compounding errors. These datasets pose varying levels of difficulty: while the expert data is relatively easier to learn from, the human and cloned datasets are much more challenging, as they require the algorithm to handle noisy, imperfect trajectories and extract meaningful learning signals from suboptimal behaviors. For the Adroit tasks, we adopt the hyperparameter configurations summarized in Table 10.

Table 10: Hyperparameter Settings for Adroit environment

| Hyper-parameter | Value |
| --- | --- |
| learning rate | 2e-4 |
| horizon | 24 |
| dropout rate | 0.1 |
| discount | 0.99 |
| codebook size | 512 |
| transformer blocks | 8 |
| attention head | 4 |
| embed dim | 512 |
| batch size | 512 |
| optimizer | Adam optimizer |

**Franka Kitchen** The Franka Kitchen environment is a multi-task, high-dimensional manipulation benchmark designed to evaluate planning and generalization in realistic, non-navigation settings. It involves controlling a 9-DoF Franka robot to interact with several common household items, including a microwave, a kettle, an overhead light, cabinets, and an oven. Each task requires the agent to manipulate these objects to reach a specific goal configuration, often involving multiple

sub-goals executed in a particular sequence. For example, a goal state may require opening the microwave and a sliding cabinet door, placing the kettle on the top burner, and turning on the overhead light. The main challenge of this environment lies in its long-horizon, sequential, and combinatorial nature. Agents must plan over multiple sub-goals while respecting their dependencies, and trajectories collected in this domain often contain complex, non-trivial paths through the state space. Success therefore depends on effective generalization to unseen states, rather than simply reproducing training trajectories.

D4RL provides three types of datasets to study this environment. The *complete* dataset consists of trajectories in which the robot performs all tasks in order, offering demonstrations that are relatively easy for imitation learning methods. The *partial* dataset contains undirected subtasks, but a subset of trajectories can still solve the tasks, allowing agents to succeed by selectively choosing relevant segments. Finally, the *mixed* dataset contains only undirected subtasks with no complete solutions, requiring agents to stitch together relevant sub-trajectories and generalize the most in order to achieve successful task execution. These datasets collectively benchmark an algorithm's ability to handle multi-task manipulation, sequential planning, and generalization in a realistic robotic environment.

Table 11: Hyperparameter Settings for Franka Kitchen environment

| Hyper-parameter | Value |
|---|---|
| learning rate | 2e-4 |
| horizon | 24 |
| dropout rate | 0.1 |
| discount | 0.99 |
| codebook size | 2048 |
| transformer blocks | 8 |
| attention head | 4 |
| embed dim | 512 |
| batch size | 512 |
| optimizer | Adam optimizer |

**Antmaze** The AntMaze benchmark is one of the most challenging tasks in offline reinforcement learning, designed to evaluate long-horizon planning and effective credit assignment under sparse reward settings. In this environment, a quadrupedal ant robot must navigate through maze-like structures to reach a distant goal location. The task is difficult due to the combination of a high-dimensional continuous action space, complex dynamics, and extremely sparse feedback, where the agent is rewarded only after successfully reaching the goal. Mazes of different sizes (U-shaped, medium, and large) also vary in complexity, with larger mazes requiring more complex planning and longer-term vision. Therefore, AntMaze is a standard platform for testing algorithms' ability to perform global reasoning and generate coherent sequences of actions with long horizons. For this environment, we adopt the hyperparameter settings summarized in Table 12.

**Maze2D & Multi2D** The Maze2D benchmark provides a series of 2D navigation tasks where a point-mass agent must traverse complex maze layouts to reach a goal position. Although the dynamics are simple, the challenge lies in long-term planning under sparse rewards, as the agent only receives a positive signal when the goal is reached. Maze2D consists of several difficulty levels, such as U-maze, medium, and large mazes, with increasing structural complexity that requires the agent to discover feasible paths across longer horizons. In this setting, the start position of the agent is fixed while the goal position is randomized across episodes, which prevents overfitting to a single target location. The Multi2D variant further increases the difficulty by randomizing both the start and goal positions in each episode, forcing the agent to generalize over a much broader distribution of navigation tasks. Together, Maze2D and Multi2D serve as canonical benchmarks for evaluating the ability of algorithms to plan effectively under sparse feedback and diverse conditions. For these environments, we adopt the hyperparameter settings reported in Table 13.

Table 12: Hyperparameter Settings for Antmaze environment

| Hyper-parameter | Value |
| --- | --- |
| learning rate | 2e-4 |
| horizon | 24 |
| dropout rate | 0.1 |
| discount | 0.998 |
| codebook size | 1024 |
| transformer blocks | 8 |
| attention head | 4 |
| embed dim | 512 |
| batch size | 512 |
| optimizer | Adam optimizer |

Table 13: Hyperparameter Settings for Maze2D and Multi2D environment

| Hyper-parameter | Value |
| --- | --- |
| learning rate | 2e-4 |
| horizon | 24 |
| dropout rate | 0.1 |
| discount | 0.99 |
| codebook size | 256 |
| transformer blocks | 4 |
| attention head | 4 |
| embed dim | 256 |
| batch size | 512 |
| optimizer | Adam optimizer |

**Gym locomotion control** The Gym locomotion control benchmark comprises a set of continuous control tasks including `HalfCheetah`, `Walker2d`, and `Ant`, where agents with articulated bodies must learn to move forward efficiently. These tasks are built on simplified physics engines that capture essential dynamics of legged locomotion. At each timestep, the agent observes a vector of physical variables such as joint angles, joint velocities, and body orientation, and outputs continuous torque commands to control its joints. The reward functions are typically dense, combining terms for forward velocity, stability, and control cost, which guide the agent toward producing smooth and sustainable gaits. Each environment poses distinct locomotion challenges: `HalfCheetah` focuses on generating rapid forward motion in a planar setting, `Walker2d` requires maintaining upright balance while walking bipedally, and `Ant` involves coordinating multiple legs to achieve stable quadrupedal movement. Collectively, these benchmarks evaluate an algorithm's ability to learn coordinated control policies under continuous dynamics and varying morphological structures. For these environments, we adopt the hyperparameter settings reported in Table 14.

## B.2 ADDITIONAL RESULTS FOR COMPARISON STUDY

In this section, we provide additional experimental results that were not included in the main text due to space limitations. These results cover a broader set of baseline methods, including flat reinforcement learning algorithms as well as several hierarchical reinforcement learning approaches.

Table 14: Hyperparameter Settings for Gym locomotion control tasks

| Hyper-parameter | Value |
|---|---|
| learning rate | 2e-4 |
| horizon | 24 |
| dropout rate | 0.1 |
| discount | 0.99 |
| codebook size | 512 |
| transformer blocks | 8 |
| attention head | 4 |
| embed dim | 512 |
| batch size | 512 |
| optimizer | Adam optimizer |

Table 15: Additional baseline comparisons for the Adroit scenarios. Results are averaged over 5 random training seeds, with each seed tested 20 times. Bold numbers indicate the best performance.

| Scenario | | BC | CQL | TT | TAP | MAGE |
|---|---|---|---|---|---|---|
| Pen | Expert | 94.6±3.2 | -1.4±2.3 | 101.8±13.8 | 127.4±7.7 | **147.8±4.9** |
| | Human | 71.0±6.2 | 13.7±16.9 | 2.0±3.4 | 76.5±8.5 | **137.1±9.0** |
| | Cloned | 51.9±15.1 | 1.0±6.6 | 38.8±13.3 | 57.4±8.7 | **108.4±17.6** |
| Door | Expert | 105.1±2.4 | -0.3±0.0 | 101.6±4.8 | 104.8±0.8 | **106.8±0.1** |
| | Human | 2.6±5.7 | 5.5±1.3 | 0.1±0.0 | 8.8±1.1 | **16.5±0.9** |
| | Cloned | -0.1±0.0 | -0.3±0.0 | 0.0±0.0 | 11.7±1.5 | **20.5±2.5** |
| Hammer | Expert | 126.7±3.8 | 0.2±0.0 | 1.1±0.2 | 127.6±1.7 | **131.7±0.2** |
| | Human | 1.2±2.7 | 0.1±0.1 | 1.4±0.1 | 1.4±0.1 | **10.4±1.2** |
| | Cloned | 0.6±0.1 | 0.3±0.0 | 0.4±0.0 | 1.2±0.1 | **13.2±4.7** |
| Recolate | Expert | 107.7±5.8 | -0.3±0.0 | 8.5±3.1 | 105.8±2.7 | **109.6±1.6** |
| | Human | 0.0±0.0 | 0.0±0.0 | 0.1±0.0 | 0.2±0.1 | **0.3±0.1** |
| | Cloned | -0.2±0.2 | -0.3±0.0 | -0.2±0.0 | -0.2±0.0 | **0.0±0.0** |
| **Average(w/o expert)** | | 15.9 | 2.5 | 5.3 | 19.6 | **38.3** |
| **Average(all settings)** | | 46.8 | 1.5 | 21.3 | 51.9 | **66.9** |

The purpose of this comparison is to offer a more comprehensive view of the performance landscape, complementing the results reported in the main paper.

**Adroit**  We additionally evaluate several representative baselines on the Adroit benchmarks. BC learns a policy by supervised imitation of expert demonstrations. CQL regularizes Q-learning to avoid overestimation of out-of-distribution actions. TT models trajectories as autoregressive sequences with a transformer. TAP (Zhang et al., 2023) leverages a discrete latent action space learned with VQ-VAE to enable efficient planning in continuous control tasks. As shown in Table 15, all these baselines are clearly outperformed by our method across different scenarios.

**Franka Kitchen**  We additionally evaluate several representative baselines on the Franka Kitchen benchmarks. BC learns a policy by supervised imitation of expert demonstrations. CQL regularizes Q-learning to avoid overestimation of out-of-distribution actions. Diffuser models trajectories as denoising diffusion processes to generate state and action sequences. As shown in Table 16, all these baselines are clearly outperformed by our method across different scenarios.

**Maze2D & Multi2D**  We further include comparisons on the Maze2D and Multi2D benchmarks with several additional baselines. MPPI (Williams et al., 2016) is a sampling-based model predictive

Table 16: Additional baseline comparisons for the Franka Kitchen Scenarios. Results are averaged over 5 random training seeds, with each seed tested 20 times. Bold numbers show the best performance.

| Scenario | | BC | CQL | Diffuser | MAGE |
|---|---|---|---|---|---|
| Kitchen | Partial | 41.3±3.7 | 51.3±7.7 | 52.5±2.5 | **91.3±3.2** |
| | Mixed | 48.9±0.7 | 51.3±7.7 | 55.7±1.3 | **86.3±3.3** |
| **Average** | | 45.1 | 51.3 | 54.1 | **88.8** |

Table 17: Additional baseline comparisons for the Maze2D and Multi2D scenarios. Results are averaged over 5 random training seeds, with each seed tested 100 times. Bold numbers indicate the best performance.

| Scenario | | MPPI | Diffuser | RGG | MAGE |
|---|---|---|---|---|---|
| Maze2D | U-maze | 33.2 | 113.9±3.1 | 108.8±1.4 | **145.4±3.2** |
| | Medium | 10.2 | 121.5±2.7 | 131.8±0.5 | **155.0±3.3** |
| | Large | 5.1 | 123.0±6.4 | 135.4±1.7 | **159.4±2.9** |
| **Single-task Average** | | 16.2 | 119.5 | 125.3 | **153.3** |
| Multi2D | U-maze | 41.2 | 128.9±1.8 | 128.3±0.8 | **150.4±1.8** |
| | Medium | 15.4 | 127.2±3.4 | 130.0±0.9 | **147.7±3.1** |
| | Large | 8.0 | 132.1±5.8 | 148.3±1.4 | **166.8±3.6** |
| **Multi-task Average** | | 21.5 | 129.4 | 135.5 | **155.0** |

Table 18: Baseline comparisons across different Gym locomotion control tasks. Bold numbers indicate the best performance.

| Task | Dataset | BC | CQL | IQL | DT | TT | CARP | TAP | Diffuser | HD | MAGE |
|---|---|---|---|---|---|---|---|---|---|---|---|
| HalfCheetah | Medium-Expert | 55.2 | 91.6 | 86.7 | 86.8 | 95.0±0.2 | 57.1±2.7 | 91.8±0.8 | 88.9±0.3 | 92.5±0.3 | **95.2±0.2** |
| | Medium | 42.6 | **49.2** | 47.4 | 42.6 | 46.9±0.4 | 38.2±1.4 | 45.0±0.1 | 42.8±0.3 | 46.7±0.2 | 43.9±0.2 |
| | Md-Replay | 36.6 | 45.5 | 44.2 | 36.6 | 41.9±2.5 | 34.6±0.6 | 40.8±0.6 | 37.7±0.5 | 38.1±0.7 | **46.0±0.2** |
| Walker2d | Medium-Expert | 107.5 | 108.8 | 109.6 | 108.1 | 101.9±6.8 | 102.2±2.8 | 107.4±0.9 | 106.9±0.2 | 107.1±0.1 | **110.3±0.1** |
| | Medium | 75.3 | 83.0 | 78.3 | 74.0 | 79.0±2.8 | 60.7±1.8 | 64.9±2.1 | 79.6±0.6 | **84.0±0.6** | 83.5±0.5 |
| | Md-Replay | 32.3 | 77.2 | 73.9 | 79.4 | 82.6±6.9 | 42.7±1.1 | 66.8±3.1 | 70.6±1.6 | 84.1±2.2 | **87.8±2.0** |
| Ant | Medium-Expert | 114.2 | 115.8 | 125.6 | 122.3 | 116.1±9.0 | 107.6±1.8 | 128.8±2.4 | 101.8±17.0 | 109.2±11.7 | **135.1±1.9** |
| | Medium | 92.1 | 90.5 | 102.3 | 94.2 | 83.1±7.3 | 76.1±0.2 | 92.0±2.4 | 79.3±10.8 | 90.1±8.9 | **107.4±1.0** |
| | Md-Replay | 89.2 | 93.9 | 88.8 | 88.7 | 77.0±6.8 | 80.2±1.2 | 96.7±1.4 | 88.1±6.2 | 83.2±1.3 | **99.3±0.9** |
| **Average** | | 71.7 | 83.9 | 84.0 | 81.4 | 80.4 | 66.6 | 81.6 | 77.3 | 81.7 | **89.8** |

control method. Diffuser leverages diffusion probabilistic models to generate trajectories for planning. RGG (Lee et al., 2023) enhances diffusion planners by introducing the recovery gap metric to detect and mitigate infeasible plans, improving both reliability and interpretability. As shown in Table 17, our method consistently achieves the best performance across both single-task and multi-task settings, outperforming all baselines.

## B.3 RESULTS FOR GYM LOCOMOTION CONTROL TASKS

We evaluate our method on standard Gym locomotion control environments, including HalfCheetah, Walker2d, and Ant. These environments involve controlling simulated agents with continuous action spaces to achieve stable and efficient locomotion. While our approach focuses on long-horizon and sparse-reward settings, it also demonstrates competitive performance on short-horizon and dense-reward tasks compared to other baselines. This indicates that our method is capable of generating coherent trajectories, maintaining fine-grained control, and effectively aligning actions with target returns across a wide range of locomotion scenarios.

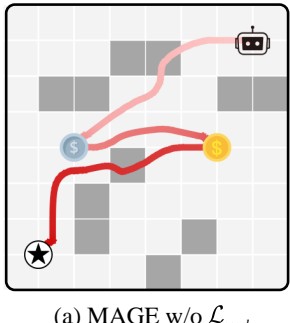 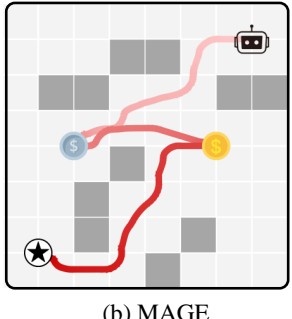

(a) MAGE w/o $\mathcal{L}_{cond}$        (b) MAGE

Figure 4: **Effect of Conditional Constraints on MAGE:** without $\mathcal{L}_{\text{cond}}$, the agent can still locate coins and reach the goal, but the agent generates the wrong trajectory that crosses the wall.

Table 19: Comparison between explicit and latent inverse dynamics models.

| Scenario | Explicit | Latent (Ours) |
|---|---|---|
| Pen-Expert | 136.5±6.7 | 147.8±4.9 |
| Door-Cloned | 0.4±0.3 | 20.5±2.5 |

Table 20: Effect of codebook size on performance.

| Scenario | 128 | 256 | 512 | 1024 |
|---|---|---|---|---|
| Pen-Expert | 112.9±19.0 | 146.5±3.1 | 147.8±4.9 | 145.3±1.9 |
| Door-Cloned | 14.4±2.9 | 19.7±3.4 | 20.5±2.5 | 11.3±2.1 |

### B.4 RESULTS FOR THE MAZE GAME IN FIGURE 2

To investigate the model's ability to capture multi-scale temporal dynamics, we designed a simple Maze game. In the game, the agent can only succeed if it starts from the initial position, collects the silver coin and the gold coin in sequence, and finally reaches the goal. In the dataset, the start and goal positions are randomized, while the silver and gold coin locations remain fixed but are not explicitly revealed by the environment. Therefore, the agent must rely on its understanding of long-horizon spatial information to identify the coin positions and navigate toward them. We study the performance of Decision Transformer, Decision Diffuser, Hierarchical Diffuser, and our method in this setting, with the results shown in Figure 1 of the main text. The results show that MAGE can finish such that long-horizon task while others cannot.

In Figure 4, we analyze the impact of conditional constraints $\mathcal{L}_{\text{cond}}$ on MAGE. Even after removing the conditional constraint $\mathcal{L}_{\text{cond}}$, MAGE still shows an understanding of the long-horizon structure, locating the silver and gold coins and reaching the goal. However, the resulting trajectories often become distorted, sometimes walking through walls. This demonstrates that our multi-scale trajectory modeling approach can effectively capture the long-horizon temporal dynamic, while the conditional constraints $\mathcal{L}_{\text{cond}}$ ensure fine-grain control and guide the model to generate more condition-compliant trajectories.

### B.5 ABLATION STUDY RESULTS

**Explicit or Latent Inverse Dynamics Model**. As illustrated in Table 19, we perform an ablation study on where to incorporate the inverse dynamics model. *Explicit* denotes decoding the trajectory first and then applying the inverse dynamics model to recover actions, and *Latent* directly decodes actions from the latent trajectory. Our approach consistently outperforms the explicit inverse model, indicating that modeling inverse dynamics in the latent space can more effectively capture trajectory-action relationships and yielding more accurate action predictions for improved planning performance.

Table 21: Effect of Transformer depth on performance.

| Scenario | 2 | 4 | 8 | 16 |
|---|---|---|---|---|
| Pen-Expert | 109.8±18.3 | 124.1±17.1 | 147.8±4.9 | 145.6±3.4 |
| Door-Cloned | 12.4±2.2 | 14.8±2.4 | 20.5±2.5 | 18.4±2.5 |

Table 22: Effect of the adapter module for applying $\mathcal{L}_{cond}$.

| Scenario | MAGE (direct decoder) | MAGE |
|---|---|---|
| Pen-Expert | 132.4±8.3 | **147.8±4.9** |
| Door-Cloned | 12.0±1.7 | **20.5±2.5** |

**Evaluating Codebook size and Transformer layers.** In this experiment, we examine two hyperparameters of our method, the size of the codebook and the number of Transformer layers. As shown in Table 20, increasing the codebook size improves performance by enriching the discrete representation, but when the codebook becomes too large the performance declines due to overfitting and excessive partitioning of the state space. Table 21 illustrates that adding more Transformer layers initially enhances performance. However, once the network reaches sufficient depth ($\geq 8$), the performance declines, indicating that further scaling provides only limited benefit.

**Role of the Adapter for Conditional Guidance.** To evaluate whether $\mathcal{L}_{cond}$ can be applied without the adapter, we introduce a variant named MAGE (direct decoder) in Table 22, where the conditional loss is added directly to the main decoder. This variant results in a noticeable performance drop, indicating that the decoder's reconstruction behavior is adversely affected. The conditional objective interferes with the learned decoding distribution, leading to degraded trajectory consistency. In contrast, the adapter cleanly separates $\mathcal{L}_{cond}$ from the decoder, ensuring stable optimization and stronger overall performance.

**Effect of RTG-Based Reweighting.** Inspired by advantage-weighted formulations such as QVPO (Ding et al., 2024), we further examine whether explicit reweighting can serve as an alternative to RTG conditioning. In our variant, each trajectory RTG is first normalized to the interval $[0, 1]$ to obtain a weight $w_i$, and the reconstruction loss becomes $w_i L_i$ instead of being averaged uniformly. We compare three settings: MAGE (w/o condition), which removes RTG conditioning entirely; MAGE (reweight), which applies the normalized RTG weights during training; and the full MAGE model. This scheme increases the influence of high-return trajectories and suppresses low-return ones. As shown in Table 23, reweighting provides a clear improvement over the no-condition baseline, indicating that it can partially leverage RTG information. However, it still falls significantly short of the full MAGE model, suggesting that direct RTG conditioning remains a more effective and reliable mechanism for guiding the generative process toward high-return behaviors.

**Evaluating performance on OGBench tasks.** We further evaluate MAGE on the extremely long-horizon and sparse-reward maze environments in OGBench (Park et al., 2025). These tasks present significant challenges due to their extended trajectories and limited feedback. As shown in Table 24, MAGE performs competitively across all scenarios and achieves the best results in three cases. While these findings demonstrate the potential of our coarse-to-fine generative framework in highly demanding settings, fully addressing such environments still requires additional exploration.

**Evaluating the default configuration.** We additionally conduct experiments using a default MAGE configuration ($K = 8$, codebook size $= 512$, transformer blocks $= 8$). As shown in Table 25, MAGE (fixed config) denotes this default setup, while MAGE refers to our final tuned model. The results show that even with the default configuration, MAGE achieves strong performance across all tasks and consistently outperforms ADT and HD.

B.6    ANALYZING THE MULTI-SCALE DESIGN AND COMPONENTS OF MAGE

Our work is a multi-scale generation method for offline RL. It consists of 4 parts: a multi-scale autoencoder, a multi-scale autoregressive transformer, an inverse dynamics model with multi-scale input, and a condition-guided decoder. The multi-scale autoencoder leverages multi-scale temporal information for encoding and decoding, whereas the multi-scale transformer leverages multi-scale

Table 23: Effect of RTG-based reweighting compared with RTG conditioning.

| Scenario | MAGE (w/o condition) | MAGE (reweight) | MAGE |
|---|---|---|---|
| Pen-Expert | $92.1 \pm 13.2$ | $131.7 \pm 10.4$ | $147.8 \pm 4.9$ |
| Door-Cloned | $0.0 \pm 0.0$ | $6.1 \pm 1.8$ | $20.5 \pm 2.5$ |

Table 24: Performance on OGBench long-horizon maze tasks.

| Scenario | Diffuser | ADT | HIQL | MAGE |
|---|---|---|---|---|
| pointmaze-giant-navigate-v0 | $0 \pm 0$ | $19 \pm 4$ | $46 \pm 9$ | $\mathbf{52 \pm 5}$ |
| antmaze-giant-navigate-v0 | $2 \pm 1$ | $26 \pm 4$ | $\mathbf{65 \pm 5}$ | $58 \pm 5$ |
| antmaze-teleport-navigate-v0 | $8 \pm 3$ | $23 \pm 4$ | $42 \pm 3$ | $\mathbf{49 \pm 5}$ |
| humanoidmaze-giant-navigate-v0 | $0 \pm 0$ | $2 \pm 1$ | $12 \pm 4$ | $\mathbf{17 \pm 4}$ |

Table 25: Performance with the default MAGE configuration.

| Scenario | ADT | HD | MAGE (fixed config) | MAGE |
|---|---|---|---|---|
| Maze2d-Medium | 109.4±6.2 | 135.6±3.0 | 146.9±4.2 | **155.0±3.3** |
| Multi2d-Medium | 108.5±6.2 | 140.2±1.6 | 142.1±1.8 | **147.7±3.1** |
| Antmaze-Medium-Play | 82.0±1.7 | 42.0±1.9 | 90.0±3.0 | **92.0±2.7** |
| Antmaze-Medium-Diverse | 83.4±1.9 | 88.7±8.1 | 94.8±2.2 | **98.2±1.3** |
| Kitchen-Mixed | 69.2±3.3 | 71.7±2.5 | 80.0±5.8 | **86.3±3.3** |
| Kitchen-Partial | 64.2±5.1 | 73.3±1.4 | 86.3±4.1 | **91.3±3.2** |

Table 26: Effect of temporal scales and the condition-guided decoder module.

| Scenario | DT | K=8 w/o $L_{cond}$ | K=1 w/o $L_{cond}$ | K=1 | K=2 | K=4 | K=8 |
|---|---|---|---|---|---|---|---|
| Pen-Expert | $116.3 \pm 1.2$ | $136.6 \pm 7.4$ | $119.5 \pm 11.5$ | $123.5 \pm 9.1$ | $127.5 \pm 5.2$ | $134.2 \pm 7.7$ | $147.8 \pm 4.9$ |
| Door-Cloned | $7.6 \pm 3.2$ | $16.6 \pm 2.1$ | $3.4 \pm 2.6$ | $5.2 \pm 1.8$ | $6.0 \pm 2.1$ | $10.7 \pm 2.3$ | $20.5 \pm 2.5$ |
| Hammer-Expert | $117.4 \pm 6.6$ | $128.3 \pm 0.3$ | $113.2 \pm 6.2$ | $116.5 \pm 0.6$ | $121.7 \pm 0.4$ | $127.9 \pm 0.3$ | $131.7 \pm 0.2$ |
| Relocate-Expert | $104.2 \pm 0.4$ | $108.9 \pm 1.7$ | $100.1 \pm 2.9$ | $101.3 \pm 3.8$ | $102.5 \pm 1.8$ | $105.9 \pm 1.4$ | $109.6 \pm 1.6$ |

temporal information for generation. The inverse dynamics model $I$ makes use of latent multi-scale trajectory information $Z$ to determine action. Moreover, the condition-guided decoder refines the finest scale information, which implicitly optimizes the multi-scale information too.

We conduct an experiment to understand the impact of multi-scale with different scales $K$ and without the condition-guided decoder (Multi-scale $w/o\ L_{cond}$). The results are depicted in the following Table 26.

In this table, the Multi-scale K=8 $w/o\ L_{cond}$ column corresponds to removing the condition-guided decoder module from MAGE. It still models the multi-scale information through a multi-scale autoencoder, a multi-scale transformer, and the inverse dynamics model with multi-scale input. Multi-scale K=1 $w/o\ L_{cond}$ is similar to Multi-scale K=8 $w/o\ L_{cond}$ with K equal to 1. $K = 1$ and $K = 8$ represent the case where K is configured to 1 or 8 for MAGE, respectively. We have the following findings.

- As we can observe from the table that using the multi-scale information can indeed performs better than its single-scale counterpart. For example, Multi-scale K=8 $w/o\ L_{cond}$ performs better than Multi-scale K=1 $w/o\ L_{cond}$, and $K = 8$ performs better than $K = 1$.

- Using the condition-guided decoder module $L_{cond}$ can improve the performance of MAGE, but its contribution is not as high as increasing the scale. For example, on the Door-Cloned environment, Multi-scale K=8 $w/o\ L_{cond}$ is 13.2 higher than Multi-scale K=1 $w/o\ L_{cond}$, while $K = 1$ (with $L_{cond}$) is only 1.8 higher than Multi-scale K=1 $w/o\ L_{cond}$.

- We show that through setting the scale to 1 and removing $L_{cond}$, MAGE performs similarly to DT. For the Door-Cloned, the Hammer-Expert, and the Relocate-Expert environment, MAGE without multi-scale and $L_{cond}$ performs even slightly weaker than Decision Transformer (DT).

## B.7 Difference among MAGE and other hierarchical methods

MAGE distinguishes itself from other hierarchical methods through its multi-level structure and trajectory modeling. While ADT, HDMI, and HD adopt a two-level hierarchy, MAGE employs a multi-level hierarchy that captures global route structures at a coarse level while refining local movements at finer levels. This design supports more coherent and consistent trajectory generation over long horizons.

The methods differ in their generation models and conditioning mechanisms. MAGE uses a Transformer-based generator to produce return–state pairs (R,S), and its condition includes the current state, target return (RTG), and outputs from higher levels. ADT and CARP generate full trajectories or action sequences, while HDMI and HD rely on diffusion models and mainly condition on state or subgoals. In this table, G, R, S, and A represent subgoal, return-to-go, state, and action, respectively Table 27 summarizes these structural and conditioning differences among the methods.

Table 27: Comparison of different methods.

|  | ADT | HDMI | HD | CARP | MAGE |
|---|---|---|---|---|---|
| Level/scales | Two-level | Two-level | Two-level | Multi-scale | Multi-scale |
| Number of Policies | 2 | 2 | 2 | 1 | 1 |
| Generation Model | Transformer | Diffusion | Diffusion | Transformer | Transformer |
| Generated Data at the first level | G | G | G | latent of A | latent of (S,R) |
| Generated Data at the last level | S,A | S,A | S,A | latent of A | latent of (S,R) |
| Condition at the first level | S | S,R | S,R | S | S,R |
| Condition at the last level | S,G | S,G | S,G | S, latents of A | (S,R), latents of (S,R) |
| Return Alignment | Yes | Yes | Yes | No | Yes |

## C Statement on the Use of Large Language Models

Large language models were used solely as general-purpose tools to assist with language refinement, including improving grammar, style, and clarity of exposition. They were not involved in generating research ideas, designing methods, conducting experiments, or analyzing results. All scientific insights and contributions presented in this paper are entirely the work of the authors.

