# OpenReview forum: "MAGE: Multi-scale Autoregressive Generation for Offline Reinforcement Learning"
_ICLR.cc/2026/Conference — ICLR 2026 Poster_

### Official Review · Reviewer_da3r · 2025-10-20

**Soundness:** 3
**Presentation:** 3
**Contribution:** 3
**Rating:** 8
**Confidence:** 3

**Summary:**

This paper presents MAGE, a novel generation-based offline reinforcement learning framework designed to tackle long-horizon tasks with sparse rewards. Unlike prior hierarchical generation methods that fail to capture the intrinsic multi-scale temporal structure of trajectories, MAGE introduces a condition-guided multi-scale autoencoder for learning hierarchical trajectory representations and a multi-scale transformer that autoregressively generates trajectories from coarse to fine temporal scales. This approach effectively models temporal dependencies across resolutions while enabling precise short-term control via a condition-guided decoder. Extensive experiments on five offline RL benchmarks against fifteen baselines demonstrate that MAGE produces coherent and controllable trajectories, achieving state-of-the-art performance in challenging long-horizon, sparse-reward settings.

**Strengths:**

- The proposed idea is novel, adapting the concept of multi-scale generation to the offline RL setting, which was originally explored in some computer vision literature (e.g., [1]).
- The empirical evaluation is comprehensive, covering a diverse set of baselines (including model-free, autoregressive, and generative approaches) and a wide variety of offline RL benchmarks such as maze navigation, MuJoCo locomotion, Adroit manipulation, and kitchen environments.
- The paper is clearly written and easy to follow, supported by intuitive illustrations (Figure 1), a well-designed toy experiment that highlights the advantages of the proposed method (Figure 2), and a clear architectural overview (Figure 3).

[1] Tian, K., Jiang, Y., Yuan, Z., Peng, B., & Wang, L. (2024). Visual autoregressive modeling: Scalable image generation via next-scale prediction. Advances in neural information processing systems, 37, 84839-84865.

**Weaknesses:**

- Since the authors claim that the proposed method performs well on long-horizon tasks, it would be interesting to evaluate it on more challenging maze tasks in OGBench [1], such as *antmaze-giant-navigate*, *antmaze-teleport-navigate*, and *humanoidmaze-giant-navigate*.
- There are a few typos in the paper — for example, “Casual” should be corrected to “Causal” in Figure 3(b).

[1] Park, S., Frans, K., Eysenbach, B., & Levine, S. (2024). Ogbench: Benchmarking offline goal-conditioned rl. arXiv preprint arXiv:2410.20092.

**Questions:**

- Could the authors provide additional results on more challenging tasks from OGBench to further demonstrate the effectiveness of the proposed method?

---

> ### Author Response · Authors · 2025-11-22
> **Response to Reviewer da3r.**
>
> Your time and effort in reviewing this work are deeply appreciated. Below, we provide detailed responses to each of your concerns.
>
> **Weakness**
>
> > W1: Since the authors claim that the proposed method performs well on long-horizon tasks, it would be interesting to evaluate it on more challenging maze tasks in OGBench [1], such as antmaze-giant-navigate, antmaze-teleport-navigate, and humanoidmaze-giant-navigate.
>
> Thank you for the suggestion. OGBench[1] is an excellent large-scale benchmark that provides a comprehensive evaluation of offline RL methods, especially in environments with extremely sparse rewards (e.g., pointmaze-giant-navigate-v0). These settings are highly challenging for offline RL due to their long horizons and extremely sparse reward signals. We compared several algorithms on four giant maze tasks. Moreover, we also compare MAGE with HIQL [2], a well-performing offline RL method for OGBench. We have discussed OGBench and HIQL in the new revision. The results are reported in the following table. We find that MAGE achieves competitive performance in these tasks.
>
> | Scenario  | Diffuser | ADT      | HIQL     | MAGE |
> |:-----------:|:-----:|:-----:|:------:|:--------:|
> | pointmaze-giant-navigate-v0 |$0 \pm 0$|$19 \pm 4$|$46 \pm 9$|$\textbf{52} \pm \textbf{5}$  |
> | antmaze-giant-navigate-v0 |$2 \pm 1$|$26 \pm 4$|$\textbf{65} \pm \textbf{5}$|$58 \pm 5$|
> | antmaze-teleport-navigate-v0 |$8 \pm 3$|$23 \pm 4$|$42 \pm 3$|$\textbf{49} \pm \textbf{5}$|
> | humanoidmaze-giant-navigate-v0 |$0 \pm 0$|$2 \pm 1$|$12 \pm 4$|$\textbf{17} \pm \textbf{4}$|
>
> Since the giant maze tasks in OGBench are very-long-horizon and extremely sparse-reward, we also study MAGE on the long-horizon Kitchen tasks, which require multi-step compositional skills. MAGE demonstrates competitive performance. The results are shown in the following table.
>
> | Scenario  | Diffuser | ADT      | HIQL     | MAGE |
> |:-----------:|:-----:|:-----:|:------:|:--------:|
> |Kitchen-Mixed|$55.7 \pm 1.3$|$69.2 \pm 3.3$|$67.7 \pm 6.8$|$\textbf{86.3} \pm \textbf{3.3}$|
> |Kitchen-Partial|$52.5 \pm 2.5$|$64.2 \pm 5.1$|$65.0 \pm 9.2$|$\textbf{91.3} \pm \textbf{3.2}$|
>
>
> These results demonstrate the effectiveness of MAGE on these challenging tasks.
>
> [1] Seohong Park, Kevin Frans, Benjamin Eysenbach, Sergey Levine, OGBench: Benchmarking Offline Goal-Conditioned RL, ICLR 25
> [2] Seohong Park, Dibya Ghosh, Benjamin Eysenbach, and Sergey Levine. HIQL: Offline goal-conditioned RL with latent states as actions. In Neural Information Processing Systems (NeurIPS), 2023
>
> > W2: There are a few typos in the paper — for example, “Casual” should be corrected to “Causal” in Figure 3(b).
>
> Thanks for the reviewer’s careful reading and feedback. We have corrected “Casual” to “Causal” in Figure 3(b) in the main paper. We will carefully proofread the paper and use a grammar-checking tool to eliminate remaining grammatical and typographical errors.
>
>
>
> **Question**
>
> > Q1: Could the authors provide additional results on more challenging tasks from OGBench to further demonstrate the effectiveness of the proposed method?
>
> OGBench is a nice benchmark for offline RL. We have already discussed the performance of MAGE in OGBench in our response to Weakness 1.

---

> > ### Comment · Reviewer_da3r · 2025-11-24
> >
> > Thank you for your response.  The additional results sound nice to me. I decided to maintain my score and recommend acceptance for this paper.
> >
> > Best regards,
> > Reviewer

---

> ### Author Response · Authors · 2025-11-25
> **Response to Reviewer da3r.**
>
> Thank you very much for your insightful review and constructive comments. We are glad that the additional results addressed your concerns, and we truly appreciate your recommendation for acceptance. We would be happy to engage in further discussion and receive any additional feedback you may have.
>
> Best regards,
>
> The authors

---

### Official Review · Reviewer_gDvQ · 2025-10-29

**Soundness:** 3
**Presentation:** 3
**Contribution:** 3
**Rating:** 4
**Confidence:** 3

**Summary:**

The paper proposes MAGE, a novel approach for offline RL by introducing a multi-scale autoencoder and a transformer to generate trajectories from coarse to fine scale. It achieves strong empirical results on several offline RL benchmarks, which require planning on long horizons with sparse rewards.

**Strengths:**

- Hierarchical generation with autoregressive models is a relevant topic in offline RL.

- Strong empirical results in several offline RL benchmarks.

- Idea is clear and also it exhibits short inference time, which is a main bottleneck of diffusion-based method for offline RL.

**Weaknesses:**

- While authors list up hyperparameters for each task in the appendix, several crucial hyperparameters are missing: RTG conditioning value, $\lambda_{\text{cond}}$, and number of temporal scales (K).

- It would be better to evaluate the method with much larger mazes, such as pointmaze-giant and antmaze-giant suggested in OGBench, to clearly verify the effectiveness of the proposed method in long-horizon settings.

**Questions:**

- It would be nice to place the overview figure in the appendix to the main text. It is hard to figure the main concept without that figure.

- It seems that with a very small number of K (e.g.,1,2), it achieves better performance compared to the baselines. Could the authors explain why such a phenomenon is possible?

---

> ### Author Response · Authors · 2025-11-22
> **Response to Reviewer gDvQ.**
>
> We greatly appreciate the time and effort you devoted to reviewing this work. Below, we address each of your concerns in detail.
>
> **Weakness**
>
> > W1: While authors list up hyperparameters for each task in the appendix, several crucial hyperparameters are missing: RTG conditioning value, $\lambda_{cond}$, and number of temporal scales (K).
>
> Thank you for pointing this out. We provide the missing hyperparameter details here. We normalize the RTG according to the task range and set the RTG to 1.0 during inference. The weight of the conditional consistency loss is $\lambda_{cond}=0.2$, and the number of temporal scales is $K=8$. We have added these details to Appendix B.1.2 in the revised paper on page 19.
>
> > W2: It would be better to evaluate the method with much larger mazes, such as pointmaze-giant and antmaze-giant suggested in OGBench, to clearly verify the effectiveness of the proposed method in long-horizon settings.
>
>
> Thank you for the suggestion. We have evaluated MAGE on the four giant mazes in OGBench and compared it with HIQL [1], an offline RL algorithm that was included in OGBench as a strong baseline/reference algorithm. We have discussed OGBench and HIQL in the new revision.
>
> The results are shown in the following table. Across the four large-scale maze tasks, **MAGE achieves the best performance on three of them**, highlighting its strong long-horizon generation capability.
>
>
> | Scenario  | Diffuser | ADT      | HIQL     | MAGE |
> |:-----------:|:-----:|:-----:|:------:|:--------:|
> | pointmaze-giant-navigate-v0 |$0 \pm 0$|$19 \pm 4$|$46 \pm 9$|$\textbf{52} \pm \textbf{5}$  |
> | antmaze-giant-navigate-v0 |$2 \pm 1$|$26 \pm 4$|$\textbf{65} \pm \textbf{5}$|$58 \pm 5$|
> | antmaze-teleport-navigate-v0 |$8 \pm 3$|$23 \pm 4$|$42 \pm 3$|$\textbf{49} \pm \textbf{5}$|
> | humanoidmaze-giant-navigate-v0 |$0 \pm 0$|$2 \pm 1$|$12 \pm 4$|$\textbf{17} \pm \textbf{4}$|
>
>
>
> Since the giant maze tasks in OGBench are very-long-horizon and extremely sparse-reward, we also study MAGE on the long-horizon Kitchen tasks, which require multi-step compositional skills. MAGE demonstrates competitive performance. The results are shown in the following table.
>
> | Scenario  | Diffuser | ADT      | HIQL     | MAGE |
> |:-----------:|:-----:|:-----:|:------:|:--------:|
> |Kitchen-Mixed|$55.7 \pm 1.3$|$69.2 \pm 3.3$|$67.7 \pm 6.8$|$\textbf{86.3} \pm \textbf{3.3}$|
> |Kitchen-Partial|$52.5 \pm 2.5$|$64.2 \pm 5.1$|$65.0 \pm 9.2$|$\textbf{91.3} \pm \textbf{3.2}$|
>
> These results demonstrate the effectiveness of MAGE on these challenging tasks.
>
> REFERENCE
> [1] Seohong Park, Dibya Ghosh, Benjamin Eysenbach, and Sergey Levine. HIQL: Offline goal-conditioned RL with latent states as actions. In Neural Information Processing Systems (NeurIPS), 2023
>
>
>
> **Questions**
>
> > Q1: It would be nice to place the overview figure in the appendix to the main text. It is hard to figure the main concept without that figure.
>
> We have moved the overview figure into the main paper on page 4, line 176.
>
>
> > Q2: It seems that with a very small number of K (e.g.,1,2), it achieves better performance compared to the baselines. Could the authors explain why such a phenomenon is possible?
>
> No, the cases of $K=1$ and $K=2$ do not lead to the best performance across all the methods studied in the submission. As shown in the following table, on Pen-Expert, with $K=1$ MAGE performs worse than IQL and TAP; with $K=2$, MAGE still performs worse than IQL. On Door-Cloned, with both $K=1$ and $K=2$, MAGE remains below DD, TAP, and DT.
>
>
> |Scenario|IQL|DD|TAP|DT|MAGE(K=1)|MAGE(K=2)|MAGE|
> |:-:|:-:|:-:|:-:|:-:|:-:|:-:|:-:|
> |Pen-Expert|$128.0 \pm 9.2$|$107.6 \pm 7.6$|$127.4 \pm 7.7$|$116.3 \pm 1.2$|$123.5 \pm 9.1$|$127.5 \pm 5.2$|$147.8 \pm 4.9$|
> |Door-Cloned|$3.0 \pm 1.7$|$9.0 \pm 1.6$|$11.7 \pm 1.5$|$7.6 \pm 3.2$|$5.2 \pm 1.8$|$6.0 \pm 2.1$|$20.5 \pm 2.5$|
>
>
> MAGE performs well even with a small $K$ because other components in our framework, such as RTG conditioning and the adapter module, also contribute to the performance. The results are shown in Table 7 of the paper.

---

> > ### Comment · Reviewer_gDvQ · 2025-11-26
> >
> > Thanks for the authors' detailed rebuttal. Most of my concerns have been resolved, especially the experiment results on OGBench and long-horizon Kitchen tasks are really nice. I have two remaining questions.
> >
> > - While $K=1$ and $K=2$ do not achieve the best performance, it seems that their relative performance is quite high. I believe that the core idea is multi-scale autoregressive generation as stated in the title, introduction, and method. However, it seems that other components are more crucial to the performance through ablation studies. Could authors verify why such phenomena happen?
> >
> > - It seems that the modified manuscript violates the page limit. It would be nice to do some modifications to fit the page limit of 9.

---

> ### Author Response · Authors · 2025-11-26
> **Response to Reviewer gDvQ.**
>
> We are happy that we have addressed most of the reviewer's concerns, especially all the weaknesses mentioned previously. We address the reviewer's new concerns in detail.
>
> >While $K=1$ and $K=2$ do not achieve the best performance, it seems that their relative performance is quite high. I believe that the core idea is multi-scale autoregressive generation as stated in the title, introduction, and method. However, it seems that other components are more crucial to the performance through ablation studies. Could authors verify why such phenomena happen?
>
>
> As we have shown in the last table of the previous response, $K=1$ and $K=2$ perform weaker than DT, DD, and TAP for the Door-Cloned task; $K=1$ and $K=2$ perform weaker than IQL for the Pen-Expert task. Their relative performance **is not quite high**. We will provide more results regarding $K=1$ and $K=2$ in the following table.
>
> Our work is a **multi-scale** generation method for offline RL. It consists of 4 parts: a multi-scale autoencoder, a multi-scale autoregressive transformer, an inverse dynamics model with multi-scale input, and a condition-guided decoder. The multi-scale autoencoder leverages multi-scale temporal information for encoding and decoding, whereas the multi-scale transformer leverages multi-scale temporal information for generation. The inverse dynamics model $I$ makes use of latent multi-scale trajectory information $Z$ to determine action. Moreover, the condition-guided decoder refines the finest scale information, which implicitly optimizes the multi-scale information through the gradient. **In total, 3.5 out of the 4 parts of our method make use of multi-scale information rather than just the multi-scale autoregressive generation part.** We have stated the term **multi-scale** clearly in the title, the abstract, the Introduction, and Section 3. We placed the term **autoregressive** in the title to emphasize that this method is an autoregressive method. We will make it clear. And we have placed such a discussion in Appendix B.6.
>
>
> We conduct an experiment to understand the impact of **multi-scale** with different scales $K$ and without the condition-guided decoder (Multi-scale $w/o\ L_{cond}$). The results are depicted in the following table.
>
>
> |Scenario|DT|Multi-scale K=8 $w/o\ L_{cond}$|Multi-scale K=1 $w/o\ L_{cond}$|K=1|K=2|K=4|K=8|
> |:-:|:-:|:-:|:-:|:-:|:-:|:-:|:-:|
> | Pen-Expert|$116.3\pm1.2$|$136.6\pm7.4$|$119.5\pm11.5$|$123.5\pm9.1$|$127.5\pm5.2$ | $134.2\pm7.7$|$147.8\pm4.9$|
> | Door-Cloned|$7.6\pm3.2$|$16.6\pm2.1$|$3.4\pm2.6$|$5.2\pm1.8$|$6.0\pm2.1$|$10.7 \pm2.3$|$20.5\pm2.5$|
> | Hammer-Expert|$117.4\pm6.6$|$128.3\pm0.3$|$113.2\pm6.2$|$116.5\pm0.6$|$121.7\pm 0.4$|$127.9\pm0.3$|$131.7\pm0.2$|
> | Relocate-Expert| $104.2\pm0.4$| $108.9 \pm 1.7$ | $100.1 \pm 2.9$ | $101.3 \pm 3.8$ | $102.5 \pm 1.8$ | $105.9 \pm 1.4$ | $109.6 \pm 1.6$|
>
>
> In this table, the Multi-scale K=8 $w/o\ L_{cond}$ column corresponds to removing the condition-guided decoder module from MAGE. It still models the multi-scale information through a multi-scale autoencoder, a multi-scale transformer, and the inverse dynamics model with multi-scale input. Multi-scale K=1 $w/o\ L_{cond}$ is similar to Multi-scale K=8 $w/o\ L_{cond}$ with K equal to 1. $K=1$ and $K=8$ represent the case where K is configured to 1 or 8 for MAGE, respectively. We have the following findings.
> * As we can observe from the table that using the multi-scale information can indeed perform better than its single-scale counterpart. For example, Multi-scale K=8 $w/o\ L_{cond}$ performs better than Multi-scale K=1 $w/o\ L_{cond}$, and $K=8$ performs better than $K=1$.
> * Using the condition-guided decoder module $L_{cond}$ can improve the performance of MAGE, **but its contribution is not as high as increasing the scale**. For example, on the Door-Cloned environment, Multi-scale K=8 $w/o\ L_{cond}$ is 13.2 higher than Multi-scale K=1 $w/o\ L_{cond}$, while $K=1$ (with $L_{cond}$) is only 1.8 higher than Multi-scale K=1 $w/o\ L_{cond}$.
> * We show that through setting the scale to 1 and removing $L_{cond}$, **MAGE performs similarly to DT**. For the Door-Cloned, the Hammer-Expert, and the Relocate-Expert environment, MAGE without multi-scale and $L_{cond}$ performs even slightly weaker than Decision Transformer (DT).
>
>
> >It seems that the modified manuscript violates the page limit. It would be nice to do some modifications to fit the page limit of 9.
>
> Thank you for the reviewer’s kind reminder regarding the page limit.
>
> According to the ICLR 2026 Author Guide
> (https://iclr.cc/Conferences/2026/AuthorGuide):
>
> *During the discussion/rebuttal phase and for the camera ready, the page limit will be increased to **10 pages** to allow for new results/discussions.*
>
> Our revised manuscript **strictly follows the page limit**. We sincerely appreciate the reviewer’s attention to formatting details and will continue to ensure that all revisions remain compliant with the official guidelines.

---

> ### Comment · Reviewer_gDvQ · 2025-11-28
>
> Thank you for the authors' clear description, and sorry for missing the guidelines for paper format. I raised my score to 6.
>
> +) It seems weird that I cannot adjust my score in openreview. Anyway, I'll update my score once the bug is resolved.

---

> ### Author Response · Authors · 2025-11-28
> **Response to Reviewer gDvQ.**
>
> Thanks very much for the increased rating (4 -> 6). We sincerely appreciate your thoughtful review and would be glad to engage in further discussion or receive any additional feedback you may have.

---

### Official Review · Reviewer_6ooy · 2025-11-01

**Soundness:** 3
**Presentation:** 3
**Contribution:** 3
**Rating:** 6
**Confidence:** 3

**Summary:**

The paper introduces MAGE, which (i) learns multi-scale discrete trajectory tokens via a condition-guided autoencoder, and (ii) performs coarse-to-fine autoregressive (AR) generation across scales with a Transformer. A lightweight adapter and an initial-condition consistency loss enforce alignment with the input condition (initial state and RTG). Discrete tokens are trained with cross-entropy and Gumbel-Softmax/STE. Experiments on long-horizon suites (Adroit, Franka Kitchen, AntMaze, Maze2D/Multi2D) show strong gains, with ablations on the number of scales, sequence schemes, and conditioning terms.

**Strengths:**

1. This paper proposes a coherent multi-scale discrete trajectory representation coupled with cross-scale AR generation (global dependencies at coarse scales; local refinements at fine scales).
2. This paper proposes a condition-guided decoder with explicit initial-condition loss to reduce quantization/AR mismatch.
3. This work presents strong empirical results on long-horizon, sparse-reward tasks, plus ablations on hierarchy depth and conditioning.

**Weaknesses:**

1. No RTG-value generalization study. The method conditions on RTG but does not report performance vs. different target RTGs (e.g., low/medium/high percentiles or out-of-distribution RTGs).
2. The same framework requires different codebook sizes, hierarchy depth K, and network depth across suites, indicating high sensitivity to task distribution and substantial tuning burden.
3. Using inverse dynamics cloning plus conditional reconstruction encourages alignment with the behaviour distribution, which can yield trajectories that “look aligned” but do not guarantee high-return action. Is it possible to use value/success reweighting?
4. The model conditions only on (s0, R) and does not explicitly encode goals or intermediate waypoints. In sparse-reward settings, if the target or waypoint distribution shifts, and especially when new destinations are not well represented in the codebook, I wonder if the performance is likely to degrade. Moreover, the coarse-to-fine AR procedure locks in global structure at coarse scales, while fine scales can only make local corrections, making early mistakes difficult to recover over long horizons.

**Questions:**

1. Related to weakness 3, is it possible to use value/success reweighting?
2. Related to weakness 4, in sparse-reward settings, if the target or waypoint distribution shifts, and especially when new destinations are not well represented in the codebook, I wonder if the performance is likely to degrade.

---

> ### Author Response · Authors · 2025-11-22
> **Response to Reviewer 6ooy. (Part1)**
>
> Thank you for your thorough review and helpful remarks. Below, we address each of your concerns in detail.
>
> **Weakness**
>
> > W1: No RTG-value generalization study. The method conditions on RTG but does not report performance vs. different target RTGs (e.g., low/medium/high percentiles or out-of-distribution RTGs).
>
>
> Thank you for this suggestion. Following the reviewer's advice, we evaluate the model's generalization to different target RTGs. In our work, we normalize the RTG and set the target RTG to 1 during inference. We evaluate the low, medium, high, and out-of-distribution (OOD) settings, corresponding to 0.3, 0.5, 1.0, and 1.2.
>
> The results are depicted in the following table. It shows that lower RTGs lead to more conservative trajectories and higher RTGs produce higher-return behaviors. Under the out-of-distribution RTG, the model still remains robust and only exhibits a slight drop in performance. These results indicate that MAGE can correctly interpret and utilize RTG signals for conditional control and also generalizes reasonably well beyond the training distribution. However, it does not address the out-of-distribution RTG case well. It is recommended to use RTG=1 only during inference for good returns.
>
>
> | Scenario | Low Percentile (0.3) | Medium Percentile (0.5) | High Percentile (1.0) | OOD (1.2) |
> |:----:|:---------:|:--------:|:------:|:------:|
> | Pen-Expert | $55.6 \pm 12.9$ | $121.6 \pm 9.9$ | $147.8 \pm 4.9$ | $135.2 \pm 8.0$ |
> | Door-Cloned | $0.0 \pm 0.0$ | $6.6 \pm 2.1$ | $20.5 \pm 2.5$ | $16.8 \pm 2.5$ |
>
> We have added these details to Appendix B.5 in the revised paper on page 24.
>
>
> > W2: The same framework requires different codebook sizes, hierarchy depth K, and network depth across suites, indicating high sensitivity to task distribution and substantial tuning burden.
>
> Thank you for the comment. A default configuration ($K = 8$, codebook size $= 512$, transformer blocks $= 8$) for MAGE performs well under multiple environments. We ran a new set of experiments using the default configuration. The following table depicts the results. We find that the default configuration provides strong and stable performance across multiple environments. It performs better than ADT and HD. Further hyperparameter tuning yields only marginal gains.
>
>
> |Scenario|ADT|HD|MAGE ($K=8$, codebook size $=512$, transformer blocks $=8$)|MAGE|
> |:-:|:-:|:-:|:-:|:-:|
> |Maze2d-Medium|$109.4 \pm 6.2$|$135.6 \pm 3.0$|$146.9 \pm 4.2$|$155.0 \pm 3.3$|
> |Multi2d-Medium|$108.5 \pm 6.2$|$140.2 \pm 1.6$|$142.1 \pm 1.8$|$147.7 \pm 3.1$|
> |Antmaze-Medium-Play|$82.0 \pm 1.7$|$42.0 \pm 1.9$|$90.0 \pm 3.0$|$92.0 \pm 2.7$|
> |Antmaze-Medium-Diverse|$83.4 \pm 1.9$|$88.7 \pm 8.1$|$94.8 \pm 2.2$|$98.2 \pm 1.3$|
> |Kitchen-Mixed|$69.2 \pm 3.3$|$71.7 \pm 2.5$|$80.0 \pm 5.8$|$86.3 \pm 3.3$|
> |Kitchen-Partial|$64.2 \pm 5.1$|$73.3 \pm 1.4$|$86.3 \pm 4.1$|$91.3 \pm 3.2$|
>
> Overall, MAGE performs reliably across different task distributions and does not rely on heavy or delicate hyperparameter tuning.

---

> > ### Author Response · Authors · 2025-11-22
> > **Response to Reviewer 6ooy. (Part2)**
> >
> > **Weakness**
> > > W3: Using inverse dynamics cloning plus conditional reconstruction encourages alignment with the behaviour distribution, which can yield trajectories that “look aligned” but do not guarantee high-return action. Is it possible to use value/success reweighting?
> >
> > Thank you for the suggestion. To better understand the reviewer's concern regarding value/success reweighting, we searched for related approaches on Google Scholar and found a representative method, QVPO[1], which performs reweighting using the advantage estimate. To evaluate reweighting, as MAGE does not learn a Q-function, we use the RTG as the reweighting weight rather than the advantage function.
> >
> > A new variant, MAGE (reweight), is introduced to address the reviewer's question about value/success reweighting. MAGE(reweight) first normalizes all trajectory RTGs and maps them to $[0,1]$ as weights $w_i$. During training, the reconstruction loss for each trajectory becomes $w_i L_i$ rather than being averaged equally $L_i$. This increases the gradient contributions from high return trajectories and suppresses the influence of low return ones.
> >
> > | Scenario | MAGE (w/o condition) | MAGE (reweight) | MAGE |
> > |:------:|:---------:|:---------:|:-----:|
> > | Pen-Expert | $92.1 \pm 13.2$ | $131.7 \pm 10.4$ | $147.8 \pm 4.9$ |
> > | Door-Cloned | $0.0 \pm 0.0$ | $6.1 \pm 1.8$ | $20.5 \pm 2.5$ |
> >
> > Here, MAGE (w/o condition) removes the RTG-based conditioning entirely, and MAGE (reweight) replaces the conditioning with RTG-based reweighting during training. The results show that reweighting alone improves performance in the absence of conditioning, but it remains noticeably below the full MAGE method. This indicates that RTG conditioned modeling is more effective at ensuring high return trajectory generation.
> >
> > We understand that value or success reweighting aims to increase the training weight of high return trajectories, thereby reducing the suboptimal bias introduced by the behavior distribution. In MAGE, however, we explicitly model the RTG $R$ and condition on RTG, which already plays a similar role to reweight. The model is naturally encouraged to learn the structure of high return trajectories during training and is explicitly guided toward the desired RTG at generation time. Empirically, this effect is clear: as shown in Table 7 of the paper and the above table, removing the RTG signal leads to a significant performance drop, indicating that this implicit "value weighting" provided by return modeling is crucial for the method's effectiveness. For convenience, we reproduce Table 7 below.
> >
> > | Scenario |MAGE ($w/o\ R\ in\ D$)| MAGE ($w/o\ R\ in\ L_{cond}$)|MAGE|
> > |:------:|:---------:|:---------:|:-----:|
> > | Pen-Expert | $140.3 \pm 9.1$ | $139.9 \pm 7.2$ | $147.8 \pm 4.9$ |
> > | Door-Cloned | $12.3 \pm 1.9$ | $17.1 \pm 2.5$ | $20.5 \pm 2.5$ |
> >
> >
> > If our implementation does not match the specific reweighting approach the reviewer refers to, we would be happy to review any related references or details the reviewer can provide.
> >
> > We have discussed [1] in the revised paper, and have added the new results to Appendix B.5 in the revised paper on page 24.
> >
> > [1] Ding, Shutong, et al. "Diffusion-based reinforcement learning via q-weighted variational policy optimization." Advances in Neural Information Processing Systems 37 (2024): 53945-53968.

---

> ### Author Response · Authors · 2025-11-22
> **Response to Reviewer 6ooy. (Part3)**
>
> **Weakness**
> > W4: The model conditions only on (s0, R) and does not explicitly encode goals or intermediate waypoints. In sparse-reward settings, if the target or waypoint distribution shifts, and especially when new destinations are not well represented in the codebook, I wonder if the performance is likely to degrade. Moreover, the coarse-to-fine AR procedure locks in global structure at coarse scales, while fine scales can only make local corrections, making early mistakes difficult to recover over long horizons.
>
> Thank you for the thoughtful observations. For unseen goals that do not appear in the dataset, this falls under the long standing OOD challenge in offline RL. We have demonstrated that OOD RTG leads to poor performance when addressing the first concern of the reviewer. It is likely that other generative approaches, including diffusion-based methods, face similar challenges in such settings.
>
> It is worth noting, however, that in the **Multi2D** experiments in Table 4, both the start and goal positions are randomly sampled at the beginning of each episode. This means the goal distribution continuously shifts throughout training and is not fixed at any specific location. Even under this setting with random and varying targets, MAGE still significantly outperforms the baselines.
>
> | Scenario | ADT | HD | MAGE |
> |:----------:|:-----:|:-----:|:--------:|
> | Multi2D-Umaze | $66.9 \pm 5.2$ | $144.1 \pm 1.2$ | $150.4 \pm 1.8$ |
> | Multi2D-Medium | $108.5 \pm 6.2$ | $140.2 \pm 1.6$ | $147.7 \pm 3.1$ |
> | Multi2D-Large | $159.4 \pm 9.2$ | $165.5 \pm 0.5$ | $166.8 \pm 3.6$ |
>
> We agree with the reviewer's second point. The coarse to fine autoregressive structure indeed fixes the global planning at coarse scales, while fine scales can only make local refinements. This design choice makes early mistakes more difficult to correct in long horizon tasks. **We have added these limitations in the discussion section of the paper on page 10.**
>
>
> **Question**
>
> > Q1: Related to weakness 3, is it possible to use value/success reweighting?
>
> We have already addressed this issue in our response to Weakness 3.
>
>
> > Q2: Related to weakness 4, in sparse-reward settings, if the target or waypoint distribution shifts, and especially when new destinations are not well represented in the codebook, I wonder if the performance is likely to degrade.
>
> We have already addressed this issue in our response to Weakness 4.

---

> ### Author Response · Authors · 2025-11-27
> **Anticipating Further Discussions with Reviewer 6ooy.**
>
> Dear Reviewer 6ooy,
>
> As the author–reviewer discussion period is ending soon, we would greatly appreciate it if you could take a moment to review our responses to your comments. If you have any further questions or feedback, we would be glad to address them while the discussion window is still open. Thank you very much for your time and consideration.
>
> Best Regards,
>
> The Authors

---

### Official Review · Reviewer_8qyC · 2025-11-09

**Soundness:** 2
**Presentation:** 3
**Contribution:** 2
**Rating:** 4
**Confidence:** 5

**Summary:**

The paper tackles challenges in offline reinforcement learning, particularly for long-horizon tasks with sparse rewards, where existing generative models struggle due to inadequate modeling of multi-scale temporal dependencies. It introduces MAGE, a method that uses a multi-scale autoencoder to encode trajectories into hierarchical token maps at varying temporal resolutions and a multi-scale transformer to autoregressively generate these maps from coarse to fine scales, conditioned on initial state and return-to-go, with a condition-guided decoder for precise trajectory control. Key contributions include integrating multi-scale trajectory modeling with conditional guidance to produce coherent trajectories. Main results from experiments on five benchmarks demonstrate superior performance against fifteen baselines, especially in sparse-reward settings, with fast inference.

**Strengths:**

- The multi-scale autoencoder captures hierarchical temporal dependencies by encoding trajectories into token maps from coarse to fine resolutions, enabling better handling of long-term structures, which enhances novelty by extending autoregressive models like VAR to temporal domains in RL.
- Figure 1 effectively contrasts MAGE's hierarchical generation with Decision Transformer's sequential and Decision Diffuser's all-at-once approaches, immediately conveying the core conceptual contribution.
- Temporal scale analysis demonstrates performance improvement with increasing K up to 8 scales, confirming the benefit of multi-scale modeling.

**Weaknesses:**

- The paper credits VAR for visual autoregressive modeling and states in the appendix that MAGE is "implemented based on the source code of... VAR". The core idea of MAGE (multi-scale, coarse-to-fine autoregressive generation) appears to be a direct application of the VAR architecture to RL trajectories. The paper should more clearly delineate its own novel contributions from the base architecture it adapts.
- The final MAGE system is highly complex, involving a multi-scale VQ-VAE, a multi-scale autoregressive transformer, a separate latent inverse dynamics model, and an "augmented decoder D' with a parameter-efficient refinement module"for the $\mathcal{L}_{cond}$ loss.
- The necessity of the latent inverse dynamics model $I(Z)$ is not fully justified. The autoencoder's decoder $\mathcal{D}$ already reconstructs the state trajectory $\hat{\tau}$. Why not decode actions as well, or infer them from consecutive reconstructed states? The ablation in Table 19 only compares a latent vs. explicit inverse model, but not the necessity of an inverse model at all.
- The introduction of a separate adapter module just to implement the $\mathcal{L}_{cond}$ loss adds complexity. It is not clear why this loss could not be applied directly to the main decoder $D$, or why this adapter-based approach is superior.
- No formal analysis or theoretical bounds explaining why the specific hierarchical factorization improves long-horizon credit assignment or reduces error propagation compared to flat models.
- The optimal number of scales K appears task-dependent but lacks principled guidance—no connection to trajectory length, reward sparsity, or environment dynamics is established.
- The condition loss L_cond (Equation 7) only constrains the initial timestep, but provides no guarantee that subsequent states respect environment dynamics or maintain trajectory coherence beyond t=0.

**Questions:**

- Could you please precisely summarize the novel algorithmic contributions of MAGE that are distinct from the VAR architecture?
- The paper states that the action is determined by a latent inverse dynamics model $a = I(Z)$. The latent representation $Z$ is defined as the set of multi-scale latents $Z = (z_1, \dots, z_K)$. How are these separate latent vectors aggregated (e.g., concatenation, summation, pooling) to form the single input $Z$ for the model $I$? This seems to be a critical implementation detail that is not specified.
- Have you performed experiments comparing MAGE (with its latent inverse model) to a simpler variant where the main decoder $\mathcal{D}$ is also trained to reconstruct actions, e.g., by modeling (R, S, A) triplets as in the ablated scheme from Table 6?
- What is the specific advantage of using a separate adapter module for this task? Why not apply the $\mathcal{L}_{cond}$ loss directly to the main decoder $\mathcal{D}$ and update its parameters? Does the adapter-based approach offer better stability, prevent catastrophic forgetting of the trajectory prior, or lead to better performance?
- Have the authors experimented with varying the number of scales K, and if so, how does performance change?
- Is there a plan to release the code or datasets upon acceptance?

---

> ### Author Response · Authors · 2025-11-22
> **Response to Reviewer 8qyC. (Part1)**
>
> Thanks for your time and effort in reviewing this work. We address your concerns as follows.
>
> **Weakness**
>
> > W1: The paper credits VAR for visual autoregressive modeling and states in the appendix that MAGE is "implemented based on the source code of... VAR". The core idea of MAGE (multi-scale, coarse-to-fine autoregressive generation) appears to be a direct application of the VAR architecture to RL trajectories. The paper should more clearly delineate its own novel contributions from the base architecture it adapts.
>
>
> Thank you for raising the question about our relationship to VAR.
> We **adapt rather than direct apply** the idea of VAR into offline RL. Our method differs from a direct VAR policy in several key aspects. We list the differences between our work (MAGE) and directly applying VAR to offline RL (VAR+RL) in the following table. VAR+RL acts as a policy to generate actions.
>
> | Method         | VAR+RL    | MAGE  |
> | - | - | - |
> | Generated Data | sequence of actions | sequence of (states, return to go) |
> | Conditioning at the First Level | $S$   | $(S, R)$             |
> | Conditioning at Later Levels | previous latents | $(S, R)$, previous latents |
> | Condition-guide Decoder | No | Yes |
> | Return Alignment | No | Yes |
> |Inverse Dynamic|No| Yes|
>
> We discuss the difference in the order of the above table.
>
> (1) VAR+RL directly generates actions, whereas MAGE generates state and action pairs.
>
> (2) In terms of formulation/condition, VAR models data as follows
> $$
> p(b_1, \dots, b_K) = \prod_{k=1}^K p(b_k \mid b_{<k}),
> $$
> Where $b_k$ denotes a spatial-scale token map in scale $k$.
>
> VAR+RL generates the first level data by using $S$ as condition, in the remaining levels of data, it does not rely on $S$ anymore.
>
> MAGE instead adopts
> $$
> p(m_1, \dots, m_K \mid s_0, R_0) = \prod_{k=1}^K p(m_k \mid m_{<k}, s_0, R_0),
> $$
> Where $m_k$ denotes a temporal-scale trajectory token map in scale $k$. In MAGE, every scale is explicitly conditioned on the initial state $s_0$, the RTG $R_0$,  and the previous latents at higher scales.
>
>
> (3) Condition-guide Decoder: When obtaining data from the last generated data of the multi-scale Transformer. MAGE introduces the conditional loss $L_{cond}$ to ensure that trajectory generation remains aligned with the intended return.
>
> (4) Return Alignment: MAGE uses RTG during the multi-scale encoding/decoding and generation process, but VAR+RL does not.
>
> (5) Inverse Dynamic: MAGE uses inverse dynamic to obtain the action, whereas VAR+RL generates actions directly.
>
> (6) We show the results among VAR+RL and MAGE across different environments in the following table. Thanks to the above improvements for RL, MAGE significantly outperforms a direct VAR policy approach.
>
> | Method | VAR + RL | MAGE |
> |:-:|:-:|:-:|
> | Pen-Expert | $112.7 \pm 19.8$ | $147.8 \pm 4.9$ |
> | Door-Cloned | $0.0 \pm 0.0$ | $20.5 \pm 2.5$ |
> | Hammer-Cloned | $0.9 \pm 0.2$ | $13.2 \pm 4.7$ |
> | Relocate-Expert | $71.0 \pm 8.3$ | $109.6 \pm 1.6$ |
> | Kitchen-Partial | $32.5 \pm 2.6$ | $91.3 \pm 3.2$ |
> | Maze2D-Medium | $65.8 \pm 2.4$ | $155.0 \pm 3.3$ |
>
>
> > W2: The final MAGE system is highly complex, involving a multi-scale VQ-VAE, a multi-scale autoregressive transformer, a separate latent inverse dynamics model, and an "augmented decoder D' with a parameter-efficient refinement module"for the $L_{cond}$ loss.
>
>
> The design of MAGE makes MAGE perform significantly better than a direct VAR+RL approach. Although MAGE contains several modules, its overall modularity is comparable to that of other generation-based offline RL methods. A component-wise comparison is shown below. Importantly, unlike hierarchical approaches that require separate high-level and low-level policies, MAGE learns only a single policy across all temporal scales.
>
> | Method | Numbers of module | modules |
> |-|-|-|
> | MAGE | 4 | multi-scale autoencoder, multi-scale autoregressive transformer, latent inverse dynamics model, adapter |
> | HD | 4 | high-level diffusion model, high-level value model, low-level diffusion model, low-level value model |
> | Decision Transformer | 3 | encoder, autoregressive transformer, decoder |
> | Decision Diffuser | 2 | diffusion, inverse dynamics model |
> VAR+RL| 2 |multi-scale autoencoder, multi-scale autoregressive transformer |
>
>
> Our multi-scale VQ-VAE and multi-scale Transformer are adapted from VAR, but our method introduces two lightweight components, the latent inverse dynamics model and the adapter module. The ablations below verify that both additions are essential.
>
> | Scenario | MAGE (w/o latent inverse dynamic) | MAGE (w/o adapter) | MAGE |
> |:-:|:-:|:-:|:-:|
> | Pen-Expert | $124.9 \pm 7.8$ | $132.4 \pm 8.3$ | $147.8 \pm 4.9$ |
> | Door-Cloned | $17.2 \pm 3.0$ | $12.0 \pm 1.7$ | $20.5 \pm 2.5$ |
>
> Here, MAGE (w/o latent inverse dynamic) refers to directly modeling the $(R, S, A)$ sequence without the latent inverse dynamics model, and MAGE (w/o adapter) denotes removing the adapter and training the decoder directly.

---

> > ### Author Response · Authors · 2025-11-22
> > **Response to Reviewer 8qyC. (Part2)**
> >
> > **Weakness**
> > > W3: The necessity of the latent inverse dynamics model $I(Z)$ is not fully justified. The autoencoder's decoder already reconstructs the state trajectory. Why not decode actions as well, or infer them from consecutive reconstructed states? The ablation in Table 19 only compares a latent vs. explicit inverse model, but not the necessity of an inverse model at all.
> >
> > The autoencoder's decoder reconstructs states and RTG, but it does not generate actions. Therefore, we use a latent inverse dynamics model $I(Z)$ to obtain actions from the generated latent trajectory. Empirically, we find that using the decoder to directly generate actions leads to a substantial performance drop. We discuss our experiments as follows.
> >
> > A new variant MAGE (direct action) is introduced. In this variant, MAGE takes $(R, S)$ as input, but the decoder is modified to directly output actions. We find that the performance drops substantially as described in the following table.
> >
> > | Scenario | MAGE (direct action) | MAGE |
> > |:----------:|:----------------:|:--------:|
> > | Pen-Expert | $126.1 \pm 10.3$ | $147.8 \pm 4.9$ |
> > | Door-Cloned | $4.7 \pm 1.7$ | $20.5 \pm 2.5$ |
> >
> > In this setting, the quantized latent codes lack a training objective that encourages the representation of the full trajectory. Their learning is mainly driven by the local action prediction task, which biases the representation toward short-horizon state–action relations rather than forming a latent structure that captures global temporal dependencies. This shift ultimately leads to clear performance degradation.
> >
> > We also evaluated two additional variants: MAGE(A) and MAGE(R, S, A) already reported in Table 6 of the initial submission. The first directly autoregresses the action sequence, and the second autoregresses the $(R, S, A)$ sequence. For the reviewer's convenience, we present them here.
> >
> > | Generated Data | MAGE(A) | MAGE(R, S, A) | MAGE |
> > |:----------:|:----:|:----------:|:--------:|
> > | Pen-Expert | $127.1 \pm 13.0$ | $124.9 \pm 7.8$ | $147.8 \pm 4.9$ |
> > | Door-Cloned | $9.1 \pm 2.3$ | $17.2 \pm 3.0$ | $20.5 \pm 2.5$ |
> >
> > >The ablation in Table 19 only compares a latent vs. explicit inverse model, but not the necessity of an inverse model at all.
> >
> > Thanks for reading our supplementary. Table 19 examines explicit inverse dynamics, which corresponds precisely to the reviewer's suggestion of inferring actions from consecutive reconstructed states, i.e., using $action = f(s_0, s_1)$. However, in high-dimensional control tasks, this approach is highly sensitive to reconstruction errors, leading to poor dynamical consistency and unstable action recovery. In contrast, the latent inverse dynamics model in MAGE directly maps latent trajectories to actions. We find that this produces smoother, more consistent, and better conditioned trajectories. The results are as follows.
> >
> > | Scenario | MAGE (Explicit) | MAGE |
> > |:----------:|:----------:|:--------:|
> > | Pen-Expert | $136.5 \pm 6.7$ | $147.8 \pm 4.9$ |
> > | Door-Cloned | $0.4 \pm 0.3$ | $20.5 \pm 2.5$ |
> >
> >
> >
> > > W4: The introduction of a separate adapter module just to implement the $L_{cond}$ loss adds complexity. It is not clear why this loss could not be applied directly to the main decoder, or why this adapter-based approach is superior.
> >
> > Thank you for raising this question. To address the reviewer's concern, we designed a new variant that directly applies $L_{cond}$ to the main decoder.
> >
> > MAGE (direct decoder) applies $L_{cond}$ directly to the main decoder during training. In this setting, the conditional objective interferes with the decoder's original trajectory reconstruction distribution, shifts its parameters, and degrades the coordination between the encoder and decoder, which consistently leads to a performance drop, as shown below.
> >
> > | Scenario | MAGE (direct decoder) | MAGE  |
> > |:----------:|:----------------:|:---------------------:|
> > | Pen-Expert | $132.4 \pm 8.3$ | $147.8 \pm 4.9$ |
> > | Door-Cloned | $12.0 \pm 1.7$ | $20.5 \pm 2.5$ |
> >
> > These results indicate that directly imposing $L_{\text{cond}}$ on the decoder harms the learned reconstruction distribution.
> >
> > To address this issue, we introduce a lightweight adapter module dedicated to optimizing $L_{cond}$. This separates the conditional objective from the main decoder, stabilizes the gradient flow, and prevents the decoder's learned generation distribution from drifting. In practice, this design yields both stable training and noticeably better performance. We have added this content to the main paper on page 9.

---

> > > ### Author Response · Authors · 2025-11-22
> > > **Response to Reviewer 8qyC. (Part3)**
> > >
> > > **Weakness**
> > > > W5: No formal analysis or theoretical bounds explaining why the specific hierarchical factorization improves long-horizon credit assignment or reduces error propagation compared to flat models.
> > >
> > > Thank you for the valuable comment. We agree with the reviewer that formal analysis can further improve this work. Our work focuses on applying multi-scale auto-regressive modeling to offline RL. The hierarchical method decomposes long horizon dependencies into shorter and more stable subproblems, where high level tokens capture global structure and low level tokens refine local details. Empirically, MAGE achieves clear gains over all flat baselines on long horizon tasks such as Maze2D and AntMaze, supporting the effectiveness of this design for long range temporal modeling.
> > >
> > > > W6: The optimal number of scales K appears task-dependent but lacks principled guidance—no connection to trajectory length, reward sparsity, or environment dynamics is established.
> > >
> > > In most environments, we use the default setting $K = 8$, and this configuration already provides strong and stable performance. To further address the reviewer's concern, we additionally evaluate different values of $K$ across multiple environments. The results are shown below:
> > >
> > > | Scenario        | 1               | 2               | 4                 | 6                 | 8                 | 10                |
> > > |:-----------------:|:-----------------:|:-----------------:|:--------------------:|:--------------------:|:--------------------:|:--------------------:|
> > > | Pen-Expert      | $123.5 \pm 9.1$ | $127.5 \pm 5.2$ | $134.2 \pm 7.7$    | $139.5 \pm 5.7$    | $147.8 \pm 4.9$    | $149.9 \pm 9.2$    |
> > > | Door-Cloned     | $5.2 \pm 1.8$   | $6.0 \pm 2.1$   | $10.7 \pm 2.3$     | $14.0 \pm 2.7$     | $20.5 \pm 2.5$     | $17.0 \pm 2.7$     |
> > > | Hammer-Expert   |$116.5 \pm 0.6$  |$121.7 \pm 0.4$  |$127.9 \pm 0.3$     |  $129.0 \pm 0.4$  | $131.7 \pm 0.2$    |$130.1 \pm 0.3$    |
> > > | Relocate-Expert | $101.3 \pm 3.8$ | $102.5 \pm 1.8$ | $105.9 \pm 1.4$    | $109.0 \pm 1.7$    | $109.6 \pm 1.6$    | $109.4 \pm 1.4$    |
> > >
> > >
> > > As shown in the table, using $K = 8$ already provides stable and strong performance, without the need for additional tuning.
> > >
> > >
> > >
> > > > W7: The condition loss L_cond (Equation 7) only constrains the initial timestep, but provides no guarantee that subsequent states respect environment dynamics or maintain trajectory coherence beyond t=0.
> > >
> > > As noted by the reviewer, $L_{cond}$ is indeed applied only at $t = 0$. In MAGE, trajectory coherence is modeled implicitly by the Multi-scale Trajectory Autoencoder and Transformer, which enforces consistent multi timestep reconstruction in latent space.
> > >
> > > Following the reviewer's advice, to further assess whether the limited constraint $L_{cond}$ at $t = 0$ affects later timesteps, we conducted experiments where the condition loss is applied at $t = 0, 1$ and across the entire trajectory. The results are as follows.
> > >
> > > | Scenario | $L_{cond} (t=0,1)$  | $L_{cond} (t=0,1,...,N)$ | MAGE |
> > > |:-----:|:--------:|:------:|:------:|
> > > | Pen-Expert | $146.0 \pm 6.9$ | $144.5 \pm 8.7$ | $147.8 \pm 4.9$ |
> > > | Door-Cloned | $18.6 \pm 3.1$ | $19.8 \pm 2.7$ | $20.5 \pm 2.5$ |
> > >
> > > The experimental results indicate that conditioning on $t=0$ can lead to better performance than condition on the whole trajectory.

---

> > > > ### Author Response · Authors · 2025-11-22
> > > > **Response to Reviewer 8qyC. (Part4)**
> > > >
> > > > **Question**
> > > > > Q1: Could you please precisely summarize the novel algorithmic contributions of MAGE that are distinct from the VAR architecture?
> > > >
> > > > We have already addressed this issue in our response to Weakness 1.
> > > >
> > > >
> > > >
> > > > > Q2: The paper states that the action is determined by a latent inverse dynamics model $a=I(Z)$. The latent representation $Z$ is defined as the set of multi-scale latents $Z=(z_1,z_2,,,z_k)$. How are these separate latent vectors aggregated (e.g., concatenation, summation, pooling) to form the single input $Z$ for the model $I$? This seems to be a critical implementation detail that is not specified.
> > > >
> > > > Thank you for pointing this out. As shown in Algorithm 2, each scale vector $z_i$ in $Z$ is obtained by first mapping the corresponding token map to a codebook embedding and then applying a linear projection to ensure a consistent dimensionality across all scales. After projection, we combine them by $Z = \sum_{i=1}^{K} z_i$. This fused latent representation $Z$ is then used as the input to the inverse dynamics model. We have updated the main text to make this clearer. We have updated this part in the main paper on page 5, line 231.
> > > >
> > > >
> > > >
> > > >
> > > > > Q3: Have you performed experiments comparing MAGE (with its latent inverse model) to a simpler variant where the main decoder $D$ is also trained to reconstruct actions, e.g., by modeling (R, S, A) triplets as in the ablated scheme from Table 6?
> > > >
> > > > We have already addressed this issue in our response to Weakness 3.
> > > >
> > > > > Q4: What is the specific advantage of using a separate adapter module for this task? Why not apply the $L_{cond}$ loss directly to the main decoder $D$ and update its parameters? Does the adapter-based approach offer better stability, prevent catastrophic forgetting of the trajectory prior, or lead to better performance?
> > > >
> > > > We have already addressed this issue in our response to Weakness 4.
> > > >
> > > > > Q5: Have the authors experimented with varying the number of scales K, and if so, how does performance change?
> > > >
> > > > Thank you for the question. We **had conducted experiments** with different numbers of scales $K$, as reported in Table 5 of the initial submission. The results show that performance generally improves as $K$ increases, and stabilizes around $K = 8$. This suggests that the multi scale structure effectively strengthens long horizon modeling, while the marginal benefit diminishes once a sufficient number of scales is reached. For the reviewer's convenience, we list the results as follows.
> > > >
> > > > | Scenario | 1 | 2 | 4 | 6 | 8 | 10 |
> > > > |:----------:|:----:|:----:|:----:|:----:|:----:|:-----:|
> > > > | Pen-Expert | $123.5 \pm 9.1$ | $127.5 \pm 5.2$ | $134.2 \pm 7.7$ | $139.5 \pm 5.7$ | $147.8 \pm 4.9$ | $149.9 \pm 9.2$ |
> > > > | Door-Cloned | $5.2 \pm 1.8$ | $6.0 \pm 2.1$ | $10.7 \pm 2.3$ | $14.0 \pm 2.7$ | $20.5 \pm 2.5$ | $17.0 \pm 2.7$ |
> > > >
> > > >
> > > >
> > > >
> > > > > Q6: Is there a plan to release the code or datasets upon acceptance?
> > > >
> > > > The code has already been included in the supplementary materials. We will release the code upon acceptance.

---

> ### Author Response · Authors · 2025-11-27
> **Anticipating Further Discussions with Reviewer 8qyC.**
>
> Dear Reviewer 8qyC,
>
> As the author–reviewer discussion period is ending soon, we would greatly appreciate it if you could take a moment to review our responses to your comments. If you have any further questions or feedback, we would be glad to address them while the discussion window is still open. Thank you very much for your time and consideration.
>
> Best Regards,
>
> The Authors

---

> ### Author Response · Authors · 2025-11-28
> **Response to Reviewer 8qyC.**
>
> Dear Reviewer 8qyC,
>
> It has been six days since we submitted our responses to your insightful comments, and we wanted to kindly follow up to see if you might have time to take another look. We truly value your feedback and would be very grateful for any further questions, suggestions, or clarifications you may have, whenever your schedule allows.
>
> Thank you again for your time and consideration.
>
> Best Regards,
>
> The Authors

---

### Meta-Review · Area_Chair_5v97 · 2026-01-08

**Summary:**

The authors propose MAGE, a multi-scale autoregressive, generation-based offline RL method that leverages hierarchical trajectory representations and coarse-to-fine generation with conditional guidance to effectively model long-horizon, sparse-reward tasks, achieving superior performance across diverse offline RL benchmarks. Although there are some flaws in the review-rebuttal discussions, I generally think the discussions are thorough and constructive. Overall, based on both the paper and the review–rebuttal discussion, I believe this work makes a meaningful and well-justified contribution to offline RL. The technical ideas are sound, the empirical results are compelling, and the paper is clearly written. I therefore recommend acceptance of this paper.

**Reviewer Concerns:**

See above comment.

**Reviewer Scores:**

Reviewer 8qyC may use LLM for language. But he/she still actively participate the rebuttal discussions, which makes think it is still OK.

---

### Decision · Program_Chairs · 2026-01-26

Accept (Poster)